# PE-SGD: Differentially Private Deep Learning via Evolution of Gradient Subspace for Text

**Tianyuan Zou[1,†], Zinan Lin[2], Sivakanth Gopi[‡], Yang Liu[3], Ya-Qin Zhang[1], Robert Sim[4], Xin Deng[4], Sergey Yekhanin[2]**

[1]Institute for AI Industry Research (AIR), Tsinghua University, [2]Microsoft Research,
[3]The Hong Kong Polytechnic University, [4]Microsoft

zty22@mails.tsinghua.edu.cn, {zinanlin,rsim,xinde,yekhanin}@microsoft.com,
sivakanth@openai.com, yang-veronica.liu@polyu.edu.hk, zhangyaqin@air.tsinghua.edu.cn
[†]Work done during an internship at Microsoft Research. [‡]Work done while working at Microsoft Research.

## Abstract

Differentially Private Stochastic Gradient Descent (DP-SGD) and its variants like DP-Adam ensure data privacy by injecting noise into per-sample gradients. Although effective with large private datasets, their performance degrades significantly when private training data is limited. Recent works leverage public data to learn a gradient subspace and project noisy private sample gradients on to this subspace, achieving improved performance. However, they have overlooked two crucial aspects: the limitation of using a fixed projection subspace throughout training and the importance of choosing where to inject noise. Therefore, we propose *Private Evolution aided Stochastic Gradient Descent* (PE-SGD), a differentially private training framework effective for scenarios with limited private data. PE-SGD uses an evolutionary strategy to update the gradient projection subspace during training process. We also identify a more effective noise injection point for better alignment between approximate DP-protected gradient and real private gradient. This enables PE-SGD to outperform DP-SGD and other baselines, particularly in the regime of limited private data and small privacy budget. Code is open-sourced at https://github.com/LindaLydia/PE-SGD.

## 1 Introduction

Large language models (LLMs) have rapidly emerged as a powerful paradigm, enabled by training on massive amounts of high-quality data (OpenAI, 2025; Comanici et al., 2025; Yang et al., 2024). Their success has reshaped natural language processing and spurred widespread adoption across industries such as finance (Iacovides et al., 2024), healthcare (Goyal et al., 2024) and personal assistants (Chew, 2022). However, this progress raises urgent concerns about training-data privacy, as LLMs are known to memorize and potentially leak sensitive information from their training corpora (Carlini et al., 2021; Lee et al., 2022; Ramakrishnan & Balaji, 2025).

Differentially private stochastic gradient descent, or DP-SGD (Abadi et al., 2016), is a classical method to address this concern by guaranteeing formal Differential Privacy (DP) (Dwork, 2006) and has recently shown great success in LLM training (McKenna et al., 2025). Its effectiveness, however, depends on sufficient training data. Due to its noisy perturbation on per-sample gradient, DP-SGD often largely underperforms vanilla non-private SGD with limited private samples (McKenna et al., 2025; Feldman et al., 2020), e.g. a few hundreds (Zou et al., 2025).

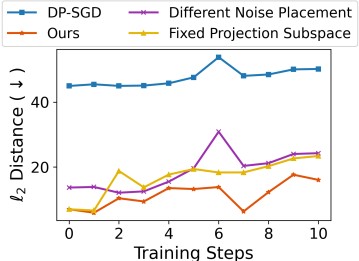

Figure 1: $\ell_2$ distance between approximate DP-protected gradient (with noise) and private gradient (without noise) when using different noise placement and fixed projection subspace compared to our proposed method, showing that both components need to be carefully selected. More results are included in Fig. 10 in Appendix D.1. See Section 1 for detail.

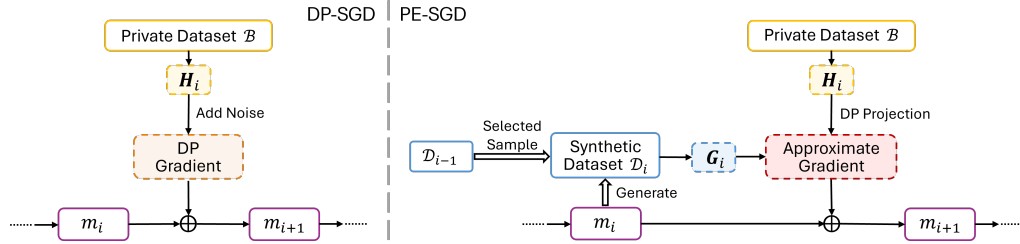

Figure 2: Overview of PE-SGD compared to standard DP-SGD (Abadi et al., 2016). Unlike DP-SGD, which updates $m$ with noisy private gradients, PE-SGD projects them into an evolutionary synthetic gradient subspace under DP and updates $m$ with the resulting approximate gradients.

Gradient projection (Yu et al., 2021; Zhou et al., 2021) has been proposed as a solution. This method addresses the challenge of high-dimensional noise in DP-SGD by projecting gradients onto a lower-dimensional subspace. This takes advantage of the empirical finding that gradient updates often lie in a much smaller subspace than the full parameter space (Gur-Ari et al., 2018; Li et al., 2020; Yu et al., 2021). The projection goal is to better approximate private gradients with non-private ones. In these works (Yu et al., 2021; Zhou et al., 2021), the eigenspace of the non-private gradients (derived from non-private samples) serves as the projection subspace.

Although this direct idea achieves better performance compared to DP-SGD, especially in the high privacy regime (tight DP budget with small $\epsilon$) on image data, two major challenges remain: **(1) Fixed projection subspace is incompatible with dynamic training.** Like shown in Fig. 1, using fixed non-private dataset ("Fixed Projection Subspace") results in increasing $\ell_2$ distance between approximate DP-protected gradient and private gradients as training progresses. This indicates that a fixed projection subspace, i.e. fixed non-private dataset, is not suitable across all training iterations, a limitation unexplored in existing literature (Zhou et al., 2021; Yu et al., 2021). Moreover, their reliance on large public dataset which itself has the concern of being sensitive (Tramèr et al., 2024). **(2) Lack of thorough exploration for the noise placement in the projection.** Existing projection methods differ in DP noise adding point. Some perturb gradient approximation coefficients (Yu et al., 2021) while others perturb private gradients (Zhou et al., 2021), or gradient residual term (Yu et al., 2021). However, the rationale behind these choices remains unclear, despite their differing effects on performance (see "Different Noise Placement" in Fig. 1).

To address these demanding challenges, we propose *Private Evolution aided Stochastic Gradient Descent* (PE-SGD), a differentially private training framework effective for scenarios with limited private data and tight privacy budget. *(1) For projection subspace that evolves during the training process*, we iteratively update the non-private dataset using the updated generative model after each parameter update. Inspired by prior DP synthetic data works (Lin et al., 2024; Xie et al., 2024), new synthetic samples are generated based on one of the selected non-private samples from current iteration using the updated model. This process starts with synthetic samples randomly generated by the (non-trained) model before training. This evolution reduces the $\ell_2$ distance between approximate DP-protected and real private gradients (see "Fixed Projection Subspace" in Fig. 1). Notably, unlike previous works (Yu et al., 2021; Zhou et al., 2021), we avoid the need for a large public dataset. *(2) For noise placement selection*, we empirically examine various options in our projection method and recommend adding the DP noise to the final projection coefficients, which yields a smaller $\ell_2$ approximation error (see "Different Noise Placement" in Fig. 1) and a significantly higher cosine similarity (see Fig. 10) between the approximate DP-protected gradient and real private gradients. Also, unlike DP-SGD, we avoid direct updates using private gradients. Our contribution are summarized as follows:

1) We propose PE-SGD, a privacy-preserving training framework robust to DP noise, particularly effective with limited private data and tight budgets (small $\epsilon$) where DP-SGD fails. Basing on a closed-form analysis identifying a principled gradient projection, we privately evolve the projection basis along with model fine-tuning for improved adaptation and empirically select a better noise injection point to better align DP-protected approximate gradients to private gradients.

2) Extensive experiments in three distinct pre-trained models and three challenging datasets (disjoint from pre-training) demonstrate the consistent superiority of PE-SGD, particularly in learning hard samples with large initial losses before DP training, achieving greater loss reductions.

## 2 RELATED WORK

**Differentially Private Training.** Differentially Private Stochastic Gradient Descent (DP-SGD) (Abadi et al., 2016) is the standard and widely applied method for training models under differential privacy (DP), achieving privacy by adding isotropic noise to sample-level gradients (McKenna et al., 2025). However, DP-SGD performs poorly with limited amount of private data compared to non-private vanilla SGD (Zhou et al., 2021). To improve utility, approaches such as Projected DP-SGD (PDP-SGD) (Zhou et al., 2021) and Gradient Embedding Perturbation (GEP) (Yu et al., 2021) have been proposed to reduce the dimension of the injected noise through gradient projection, while Sparsified Model Perturbation (SMP) (Hu et al., 2023) sparsifies model parameters directly. Although proven effective, a principled gradient projection method remains unrevealed.

**Differentially Private Synthetic Data.** With the rise of generative models such as Large Language Models (OpenAI, 2025; Yang et al., 2025) and Stable Diffusion (Rombach et al., 2022) recently, synthetic data has attracted increasing attention as an alternative in scenarios where large-scale, high-quality real data is scarce or sensitive. However, the long-recognized low-quality issue of synthetic data (Ye et al., 2022; Gao et al., 2023; Zou et al., 2024) limits its direct use for model training or fine-tuning. To address this, private domain data has been used as generation guidance to align synthetic data distributions. Yet, traditional DP-SGD–based methods (Bommasani et al., 2019; Mattern et al., 2022; Putta et al., 2023) are often impractical in resource-constrained settings (Abou Baker et al., 2024; Wornow et al., 2023) and infeasible with powerful closed-source generative models (Xie et al., 2024; Hou et al., 2024). More recently, Private Evolution (PE) (Lin et al., 2024) and its variants (Xie et al., 2024; Hou et al., 2024; Zou et al., 2025; Lin et al., 2025; Wang et al., 2025b) refine synthetic data to resemble private domain data in the embedding space using only generative APIs. However, they do not directly optimize samples for training or fine-tuning.

## 3 PRELIMINARIES

**Differential Privacy (DP).** Two datasets $\mathcal{D}$ and $\mathcal{D}'$ are referred to as *Neighboring Datasets* ($\mathcal{D} \sim \mathcal{D}'$) if they differ in a single entry. If $\Pr[\mathcal{M}(\mathcal{D}) \in E] \leq e^\epsilon \cdot \Pr[\mathcal{M}(\mathcal{D}') \in E] + \delta$ holds for any neighboring datasets $\mathcal{D}, \mathcal{D}'$ and any output subset $E$ of a random mechanism $\mathcal{M}$, then $\mathcal{M}$ satisfies $(\epsilon, \delta)$-DP (Dwork, 2006). Note that any post-processing on the output of $(\epsilon, \delta)$-DP does not incur additional privacy loss (Dwork et al., 2014).

**Gaussian Mechanism.** For any $\epsilon > 0, \delta \in (0, 1)$, following Gaussian Mechanism (Dwork, 2006), $(\epsilon, \delta)$-DP can be satisfied by adding Gaussian noise following $\mathcal{N}(0, \sigma^2)$ where the noise level $\sigma$ is decided on the basis of $\epsilon, \delta$. We use the PRV Accountant from Gopi et al. (2021) for the calculation of $\sigma$ in our work.

## 4 METHODOLOGY

### 4.1 PROBLEM DEFINITION

In this paper, our goal is to train a generative model $m$ that better adapts to the user-selected private distribution (task) from a pre-trained generative model $m^{(0)}$. To be specific, given a private dataset $\mathcal{B}$ of size $M$, which at the same time defines a task, the final goal is to fine-tune a pre-trained model $m^{(0)}$ for learning the private dataset distribution while guaranteeing $(\epsilon, \delta)$-DP of $\mathcal{B}$. In this paper, we mainly consider a limited private data setting, with $M < 500$ following Zou et al. (2025). The fine-tuned model $m$ is evaluated on a private dataset $\mathcal{A}$ reserved for evaluation only.

### 4.2 OUR METHOD PE-SGD

**1) Gradient Projection.** Since DP-SGD (Abadi et al., 2016) suffers from severe performance degradation with limited private samples, gradient projection methods (Zhou et al., 2021; Yu et al., 2021) have been proposed to reduce the noise dimension. The core idea is to approximate private gradients $\boldsymbol{H} \in \mathbb{R}^{p \times M}$ under DP protection using non-private gradients $\boldsymbol{G} \in \mathbb{R}^{p \times N}$ of a non-private dataset $\mathcal{D}$ of size $N$, with $p$ being the number of trainable parameters of current model $m$.

Given a set of private gradients $\boldsymbol{H} = [\boldsymbol{h}_1, \ldots, \boldsymbol{h}_M] \in \mathbb{R}^{p \times M}$, non-private SGD updates the model with their average $\boldsymbol{h}$. Therefore, with a set of synthetic gradients $\boldsymbol{G} = [\boldsymbol{g}_1, \ldots, \boldsymbol{g}_N] \in \mathbb{R}^{p \times N}$ serving as the projection subspace, our goal is to minimize the $\ell_2$ distance between the approximation and $\boldsymbol{h}$. This can be formulated as:

$$\min_{\boldsymbol{z} \in \mathbb{R}^N} ||\boldsymbol{G}\boldsymbol{z} - \boldsymbol{h}||_2^2, \text{ with } \boldsymbol{h} = \frac{1}{M} \sum_{j=1}^M \boldsymbol{H}_{:,j} = \frac{1}{M} \sum_{j=1}^M \boldsymbol{h}_j, \tag{1}$$

where $\boldsymbol{z} \in \mathbb{R}^N$ is the coefficient vector. Solving Eq. (1) is equivalent to least-squares regression in the under-parametrized regime (since $N \ll p$) where we can assume that the $N \times N$ matrix $\boldsymbol{G}^T\boldsymbol{G}$ is invertible.[1] Therefore, the least-squares solution (which can be obtained by setting the gradient of objective in Eq. (1) to zero) is given by:

$$\boldsymbol{z} = (\boldsymbol{G}^T\boldsymbol{G})^{-1}(\boldsymbol{G}^T\boldsymbol{h}) = \frac{1}{M} \sum_{j=1}^M (\boldsymbol{G}^T\boldsymbol{G})^{-1}(\boldsymbol{G}^T\boldsymbol{h}_j) = \frac{1}{M} \sum_{j=1}^M [(\boldsymbol{G}^T\boldsymbol{G})^{-1}(\boldsymbol{G}^T\boldsymbol{H})]_{:,j}. \tag{2}$$

The final projection function can therefore be written as:

$$\hat{\boldsymbol{g}} = \boldsymbol{G}\boldsymbol{z} = \frac{1}{M} \sum_{j=1}^M [\boldsymbol{G}(\boldsymbol{G}^T\boldsymbol{G})^{-1}(\boldsymbol{G}^T\boldsymbol{H})]_{:,j}. \tag{3}$$

Current method PDP-SGD (Zhou et al., 2021) instead proposed to project as $\hat{\boldsymbol{g}} = \boldsymbol{E}\boldsymbol{E}^T(\sum_{j=1}^M \boldsymbol{H}_{:,j})$ with $\boldsymbol{E} \in \mathbb{R}^{p \times k}$ spans the top-$k$ eigenspace of $\boldsymbol{G}\boldsymbol{G}^T$ while GEP defines $\boldsymbol{E}$ as the $k$ components of the left singular vectors corresponds to the top-$k$ singular values of $\boldsymbol{G}$ and additionally adds a residual term $\boldsymbol{R} = \boldsymbol{H} - \boldsymbol{E}\boldsymbol{E}^T\boldsymbol{H}$ to better correct the bias resulting in $\hat{\boldsymbol{g}} = \boldsymbol{E}\boldsymbol{E}^T(\sum_{j=1}^M \boldsymbol{H}_{:,j}) + \sum_{k=1}^M \boldsymbol{R}_{:,j}$. Note that these gradient projection methods are equal to ours given in Eq. (3), if it does not use the top-$k$ eigenspace but the whole space. Proof is included in Appendix B.3. Intuitively, however, restricting to a subspace inevitably incurs a performance loss.

**2) Existing Key Challenges.** With the principled gradient projection given in Eq. (3), several key challenges still exist.

*[C1] How to improve fixed datasets for better gradient projections?* Beyond projection design, using a proper non-private dataset $\mathcal{D}$ for the projection subspace $\boldsymbol{G}$ is also crucial. Intuitively, if $\mathcal{D}$ differs greatly from the private dataset $\mathcal{B}$, e.g. $\mathcal{D}$ is non-overlapping with $\mathcal{B}$ in distribution, the model cannot effectively learn the knowledge in $\mathcal{B}$, which could lead to poor training results. Prior works (Yu et al., 2021; Zhou et al., 2021) rely on a fixed (public) dataset as $\mathcal{D}$ for the projection subspace $\boldsymbol{G}$. But as training progresses, this fixed base causes the $\ell_2$ distance between the real private gradient and the approximate DP-protected gradient to grow (see "Fixed Projection Subspace" in Fig. 1).

*[C2] Where should the DP noise be integrated in the projection?* Another interesting unresolved issue is the injection place of the random noise for DP guarantee. DP-SGD (Abadi et al., 2016) and PDP-SGD (Zhou et al., 2021) added noise (of shape $\mathbb{R}^p$) to aggregated per-sample private gradients $\frac{1}{M} \sum_{j=1}^M \boldsymbol{H}_{:,j}$ after clipping; while there exists other works like GEP (Yu et al., 2021) (and its follow up Gu et al. (2025)) that adds noise (of shape $\mathbb{R}^N$) after projection in the projection space. However, the rationale behind these noise placement selection remains unclear, while there exist multiple possible noise injection places within the gradient projection process in Eq. (2).

**3) Our solutions.** We address these two key challenges that elucidate our proposed method, `PE-SGD`.

*[S1] Synthetic Dataset Evolution.* Unlike prior work that relies on a fixed $\mathcal{D}$ (Zhou et al., 2021; Yu et al., 2021), we construct $\mathcal{D}$ that evolves during fine-tuning to adapt to the updated model $m$. Inspired by PE (Lin et al., 2024; Xie et al., 2024), and starting from a pre-trained generator $m^{(0)}$ with its randomly generated synthetic samples (termed as `RANDOM_API()` in Xie et al. (2024)), we prompt each updated $m$ to produce new samples for the next iteration by varying carefully selected seed samples $\mathcal{D}^{seed}$. At each iteration, each element $z_i$ in $\boldsymbol{z} \in \mathbb{R}^N$ can be seen as a score of synthetic

---

[1]In practice, to ensure numerical stability, we add a small $\ell_2$ regularization term to the least-squares problem. Thus we invert $(\boldsymbol{G}^T\boldsymbol{G} + \eta I)$ with $\eta = 10^{-6}$ instead.

---

**Algorithm 1** `PE-SGD`

---

**Input:** Pre-trained generative model $m^{(0)}$ with parameter $\phi$; empty synthetic dataset $\mathcal{D} \leftarrow \emptyset$; private dataset $\mathcal{B}$ of size $M$; number of iterations for updates $T$; number of synthetic samples to generate $N$ per iteration; number of synthetic sample variation fold $L$; Poisson sub-sampling ratio $\beta$ for private samples; learning rate $\eta$; DP parameters $\epsilon, \delta$;
**Output:** Fine-tuned or trained PLM $m$.

1: $\mathcal{D}_{seed}^{(0)} \leftarrow \emptyset, \mathcal{D} \leftarrow$ SynDataGeneration( $\mathcal{D}_{seed}^{(0)}, N, L, m^{(0)}$ ).
2: $m \leftarrow$ fine-tuned $m^{(0)}$ using $\mathcal{D}$.
3: $\sigma \leftarrow$ NoiseCalculation( $\epsilon, \delta, T, \beta$ ) following Gopi et al. (2021).
4: **for** $t = 1$ **to** $T$ **do**
5: $\quad \mathcal{B}^{(t)} \leftarrow$ PoissonSubSampling( $\mathcal{B}, \beta$ ), $\tilde{M} \leftarrow |\mathcal{B}^{(t)}|$.
6: $\quad \boldsymbol{G} \leftarrow [\boldsymbol{g}_1, \ldots, \boldsymbol{g}_N]$ with $\boldsymbol{g}_i$ being the sample-level gradient of $\mathbf{x}_i \in \mathcal{D}$ on $m$.
7: $\quad \boldsymbol{H} \leftarrow [\boldsymbol{h}_1, \ldots, \boldsymbol{h}_{\tilde{M}}]$ with $\boldsymbol{h}_j$ being the sample-level gradient of $\mathbf{w}_j \in \mathcal{B}^{(t)}$ on $m$.
8: $\quad \boldsymbol{z} \leftarrow$ SumUpNormalizedColumns( $(\boldsymbol{G}^T\boldsymbol{G})^{-1}(\boldsymbol{G}^T\boldsymbol{H})$ ) $+ \mathcal{N}(0, \sigma^2 I_N)$.
9: $\quad \phi \leftarrow \phi + \eta \cdot [(\boldsymbol{G}\boldsymbol{z})/\tilde{M}]$ to update $m$.
10: $\quad \mathcal{D}_{seed}^{(t)} \leftarrow$ SampleSelection( $\boldsymbol{z}, N, L, \mathcal{D}$ ), $\mathcal{D} \leftarrow$ SynDataGeneration( $\mathcal{D}_{seed}^{(t)}, N, L, m$ ).
11: **end for**

---

sample $\mathbf{x}_i$ given by private samples. We therefore select top-$K$ seed samples with probability proportional to $|z_i|$, since gradient related information can also be extracted from negative $z_i$. We then expand each seed sample into $L-1$ variants via a one-shot prompt (termed as `VARIATIONAL_API()` in Xie et al. (2024)). Gaussian noise $\sim \mathcal{N}(0, \sigma^2 I_N)$ is added to $\boldsymbol{z}$ prior to sample selection for privacy consideration.[2] Detailed prompts for generation are included in Appendix C.2. Note this evolutionary design removes the need for large public datasets. Moreover, with synthetic dataset of small size (200 in our experiments) serving as the projection subspace, we further avoid the need of using top-$k$ eigenspace and exploits the entire projection subspace for gradient approximation.

***[S2] Systematic Study of the DP Noise Addition Place.*** Prior works select their own way of adding noise arbitrarily. However, from Eq. (2), we identify three natural options: (1) adding noise of shape $\mathbb{R}^p$ to the aggregated private gradient $\sum_{i=1}^{M} \boldsymbol{H}_{:,j}$, (2) adding noise of shape $\mathbb{R}^N$ to the gradient inner product $\sum_{j=1}^{M}[\boldsymbol{G}^T\boldsymbol{H}]_{:,j}$, or (3) adding noise of shape $\mathbb{R}^N$ to the final projection coefficient $\sum_{j=1}^{M}[(\boldsymbol{G}^T\boldsymbol{G})^{-1}(\boldsymbol{G}^T\boldsymbol{H})]_{:,j}$. Our empirical study (Fig. 8) shows that the third option consistently yields the best performance.

**4) Feasibility Discussion** Naively computing $\boldsymbol{G}^T\boldsymbol{G}$ and $\boldsymbol{G}^T\boldsymbol{H}$ with $\boldsymbol{G} \in \mathbb{R}^{p \times N}, \boldsymbol{H} \in \mathbb{R}^{p \times \tilde{M}}$ in `PE-SGD` in Algorithm 1 can look infeasible for large models since all the per-sample gradients do not even fit in the GPU memory, so we cannot materialize the matrix $\boldsymbol{G}$ or $\boldsymbol{H}$. There are two ways to get around this: (1) We can use parameter efficient fine-tuning methods like LoRA (this is the approach we take in this paper) which makes $p$ relatively small. Therefore, per-sample gradient matrices $\boldsymbol{G}, \boldsymbol{H}$ and the rest of the matrix multiplications can be computed directly. (2) If we want to do full fine-tuning, the per-sample gradient dot-products $\boldsymbol{G}^T\boldsymbol{G}$ and $\boldsymbol{G}^T\boldsymbol{H}$ can be efficiently obtained using GhostSuite (Wang et al., 2025a),[3] which computes gradient dot-products between all pairs of data points in a large batch with a single backpropagation.

**5) Overall Framework** We summarize our `PE-SGD` algorithm in Fig. 2 and Algorithm 1. Functions applied in Algorithm 1 are included in Algorithm 3 in Appendix B.1. Notice that normalization instead of clipping is used in Algorithm 1 for sensitivity control before column aggregation, as both techniques yield similar results while normalization avoids tuning the hyper-parameter clipping threshold $C$ used in clipping. Also, we use the same sub-sampling strategy as DP-SGD, ensuring the only difference lies in how the sub-sampled private batch $\mathcal{B}^{(t)}$ for each iteration $t$ is utilized. A full privacy analysis of `PE-SGD` is provided in Appendix B.2. Last but not least, although Algorithm 1 is presented in the form of vanilla SGD (line 9), it is fully compatible with advanced optimizers such as AdamW, simply by feeding our approximate gradients into them. A mathematical convergence analysis of `PE-SGD` compared to previous methods are included in Appendix E showing that the convergence speed of `PE-SGD` is no slower than that of previous methods.

---

[2]We directly follow Gopi et al. (2021) for the calculation of the noise scale $\sigma$.
[3]https://github.com/Jiachen-T-Wang/GhostSuite

# 5 Experimental Results

Table 1: Performance comparison of the trained model $m$ using PE-SGD and baseline methods. **Best** and second best results (except SGD and DP-SGD) are marked.

| Method | PubMed | | Congressional Speech | | bioRxiv | |
|---|---|---|---|---|---|---|
| | $\varepsilon = 1$ | $\varepsilon = \infty$ | $\varepsilon = 1$ | $\varepsilon = \infty$ | $\varepsilon = 1$ | $\varepsilon = \infty$ |
| **Next Token Prediction Loss ($\downarrow$)** | | | | | | |
| Base Model | 2.9341 | | 3.8690 | | 2.5311 | |
| SGD | $2.1197_{\pm 0.0004}$ | | $2.6078_{\pm 0.0084}$ | | $2.2756_{\pm 0.0001}$ | |
| DP-SGD | $2.3731_{\pm 0.0220}$ | $2.1497_{\pm 0.0037}$ | $3.1080_{\pm 0.0081}$ | $2.6868_{\pm 0.0383}$ | $2.3768_{\pm 0.0041}$ | $2.2887_{\pm 0.0021}$ |
| Aug-PE | $2.5817_{\pm 0.1537}$ | $2.6070_{\pm 0.2190}$ | $3.0090_{\pm 0.1063}$ | $3.0120_{\pm 0.1048}$ | $2.4672_{\pm 0.0169}$ | $2.4672_{\pm 0.0169}$ |
| POPri | $2.9344_{\pm 0.0023}$ | $2.9487_{\pm 0.0208}$ | $3.8656_{\pm 0.0082}$ | $3.8700_{\pm 0.0038}$ | $2.6178_{\pm 0.0260}$ | $2.5309_{\pm 0.0083}$ |
| PDP-SGD | $2.2502_{\pm 0.0370}$ | $2.2482_{\pm 0.0414}$ | $2.8025_{\pm 0.0128}$ | $2.8000_{\pm 0.0128}$ | $2.3325_{\pm 0.0032}$ | $2.3307_{\pm 0.0011}$ |
| GEP | $2.4164_{\pm 0.0625}$ | $2.2248_{\pm 0.0141}$ | $2.9103_{\pm 0.0263}$ | $2.7822_{\pm 0.1470}$ | $2.4300_{\pm 0.0058}$ | $2.3199_{\pm 0.0029}$ |
| InPro | $2.5268_{\pm 0.0543}$ | $2.5229_{\pm 0.0625}$ | $3.4693_{\pm 0.1381}$ | $3.4728_{\pm 0.1433}$ | $2.4079_{\pm 0.0238}$ | $2.4081_{\pm 0.0236}$ |
| PE-SGD-FixSample | $2.2497_{\pm 0.0091}$ | $2.2462_{\pm 0.0047}$ | $2.8024_{\pm 0.0535}$ | $2.8137_{\pm 0.0167}$ | $2.3230_{\pm 0.0009}$ | $2.3240_{\pm 0.0062}$ |
| PE-SGD | **$2.1990_{\pm 0.0124}$** | **$2.1839_{\pm 0.0131}$** | **$2.7532_{\pm 0.0166}$** | **$2.7468_{\pm 0.0095}$** | **$2.3178_{\pm 0.0018}$** | **$2.3165_{\pm 0.0040}$** |
| **Next Token Prediction Accuracy (ACC) ($\uparrow$)** | | | | | | |
| Base Model | 0.5217 | | 0.4276 | | 0.4981 | |
| SGD | $0.5494_{\pm 0.0004}$ | | $0.4661_{\pm 0.0013}$ | | $0.5145_{\pm 0.0004}$ | |
| DP-SGD | $0.5405_{\pm 0.0005}$ | $0.5461_{\pm 0.0003}$ | $0.4492_{\pm 0.0005}$ | $0.4656_{\pm 0.0009}$ | $0.5075_{\pm 0.0004}$ | $0.5130_{\pm 0.0005}$ |
| Aug-PE | $0.5203_{\pm 0.0131}$ | $0.5201_{\pm 0.0146}$ | $0.4450_{\pm 0.0110}$ | $0.4447_{\pm 0.0113}$ | $0.4982_{\pm 0.0015}$ | $0.4982_{\pm 0.0015}$ |
| POPri | $0.5221_{\pm 0.0013}$ | $0.5192_{\pm 0.0039}$ | $0.4276_{\pm 0.0082}$ | $0.4276_{\pm 0.0018}$ | $0.4930_{\pm 0.0028}$ | $0.4981_{\pm 0.0035}$ |
| PDP-SGD | $0.5382_{\pm 0.0002}$ | $0.5385_{\pm 0.0005}$ | $0.4546_{\pm 0.0008}$ | $0.4566_{\pm 0.0009}$ | $0.5092_{\pm 0.0002}$ | $0.5093_{\pm 0.0001}$ |
| GEP | $0.5344_{\pm 0.0013}$ | $0.5423_{\pm 0.0010}$ | $0.4549_{\pm 0.0007}$ | $0.4566_{\pm 0.0020}$ | $0.5035_{\pm 0.0004}$ | $0.5095_{\pm 0.0003}$ |
| InPro | $0.5316_{\pm 0.0015}$ | $0.5315_{\pm 0.0015}$ | $0.4331_{\pm 0.0027}$ | $0.4330_{\pm 0.0015}$ | $0.5045_{\pm 0.0016}$ | $0.5045_{\pm 0.0016}$ |
| PE-SGD-FixSample | $0.5413_{\pm 0.0007}$ | $0.5420_{\pm 0.0010}$ | $0.4548_{\pm 0.0035}$ | $0.4561_{\pm 0.0024}$ | $0.5092_{\pm 0.0003}$ | $0.5095_{\pm 0.0005}$ |
| PE-SGD | **$0.5435_{\pm 0.0009}$** | **$0.5436_{\pm 0.0005}$** | **$0.4577_{\pm 0.0011}$** | **$0.4579_{\pm 0.0021}$** | **$0.5095_{\pm 0.0000}$** | **$0.5097_{\pm 0.0004}$** |

## 5.1 Experimental Settings

**Pre-trained Language Models.** Three pre-trained generative language models are used in experiments: Qwen2.5-3B-Instruct (Qwen2.5) (Qwen-Team, 2024) released on September $19^{th}$, 2024; Llama-3.2-3B-Instruct (Llama3.2) (Dubey et al., 2024) released on September $25^{th}$, 2024; and GPT2 (Radford et al., 2019) released on November $5^{th}$, 2019. We use instruction fine-tuning in our experiments for the first 2 models as they are instruction-fine-tuned models.

**Tasks and Datasets.** Considering the release date of the above models, we carefully select real-world data samples (used as private samples in both training dataset $\mathcal{B}$ and evaluation dataset $\mathcal{A}$) that are created after the model release dates to ensure that these models have not seen them. To be specific, three datasets are used in our experiments: 1) PubMed (NLM (U.S.), 2025) dataset containing abstracts of professional biomedical literature completed from December 2024 to January 2025; 2) Congressional Speech (Hou et al., 2025) dataset containing public speeches extracted from government institutions of the US, UK and Canada from January 2025 to June 2025; 3) bioRxiv (Hou et al., 2025) dataset containing abstracts of articles on bioRxiv website, a pre-printing server for biology created from January 2025 to June 2025. More details are included in Appendix C.1.

**Baselines.** In this work, we compare with five baselines: 1) DP-SGD (Abadi et al., 2016) which is the basic DP training method that is widely applied; 2) Aug-PE (Xie et al., 2024) which generates DP synthetic data that are further exploited for (normal, non-DP) supervised fine-tuning; 3) POPri (Hou et al., 2025) which exploits Direct Preference Optimization (DPO) for fine-tuning on DP synthetic data; 4) PDP-SGD (Zhou et al., 2021); 5) GEP (Yu et al., 2021). We also compare with "InPro" which adopts a biased projection method using gradient inner products with $\hat{g} = \frac{1}{M} \sum_{j=1}^{T} G[G^T H]_{:,j}$. We further compare with 1 important variant or PE-SGD, namely "PE-SGD-FixSample" that performs the same Algorithm 1 but without updating $\mathcal{D}$ (line 10), to demonstrate the power of synthetic sample evolution.

**Evaluation Metrics.** As our goal is to fine-tune a generative model that better adapts to the private data domain, we use the most commonly applied evaluation metrics in language model training as

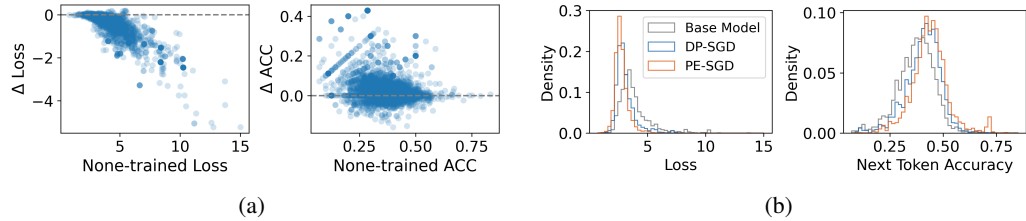

(a)                                 (b)

Figure 3: $(a)$ Per-sample loss and accuracy (ACC) differences between PE-SGD and DP-SGD with each point corresponding to one sample. The x-axis shows the initial loss on $m^{(0)}$, and the y-axis shows the per-sample loss or accuracy difference after training (PE-SGD minus DP-SGD). $(b)$ Loss and accuracy (ACC) distribution change before and after training with PE-SGD and DP-SGD. Qwen2.5 is used for Congressional Speech.

Table 2: Average loss change of the top $10\%$ samples with the largest initial loss on base model $m^{(0)}$ after fine-tuning using different methods. Qwen is used for Congressional Speech.

| | PE-SGD | PE-SGD-FixSample | Baselines | | | | | |
| --- | --- | --- | --- | --- | --- | --- | --- | --- |
| | | | PDP-SGD | GEP | DP-SGD | Aug-PE | InPro | POPri |
| $\Delta$ Loss | -3.49995 | -3.41537 | -3.32709 | -3.01130 | -2.03666 | -1.67257 | -1.28827 | -0.00034 |

our evaluation metrics, namely, next token prediction loss and accuracy. We refer to them as Loss and Accuracy (or ACC) respectively in the following sections for brevity.

**Learning Hyper-parameters.** Unless otherwise stated, we use $M = 400, \beta = 0.2, T = 10, N = 200, L = 2$ and guarantee $(1.0, 1e^{-5})$-DP for LoRA fine-tuning in our experiments. More detailed hyper-parameter settings are included in Appendix C.3.

## 5.2 MAIN RESULTS

We first comprehensively evaluate the loss and accuracy of the fine-tuned generative model $m$ on the evaluation private dataset $\mathcal{A}$ in Table 1. Note when $\epsilon = \infty$, DP is not guaranteed as $\sigma = 0.0$, whereas norm clipping or normalization is still performed respectively. Results show that PE-SGD outperforms all other methods when $\epsilon = 1.0$, achieving lower loss and higher accuracy, indicating that $m$ learns more about the private distribution while preserving data privacy under tight DP budget. For $\epsilon = \infty$, it's expected that DP-SGD achieves better results as it directly trains on norm-clipped private samples gradients. On the other hand, "PE-SGD-FixSample" is also competitive compared to most baselines, making it a good substitute under resource limited settings, as it saves the cost of iteratively updating the synthetic dataset $\mathcal{D}$.

POPri does not exhibit good performance under small private training dataset setting ($M = 400$), as it is hard to obtain reliable positive-negative sample pairs with limited private data. Aug-PE also performs unsatisfactory under this small private data setting, similar to that shown in Zou et al. (2025). Other gradient projection methods do not guarantee a consistent better performance than the basic baseline DP-SGD, testifying that their projection method is suboptimal (InPro) or has some information loss when using only an eigen sub-space (PDP-SGD and GEP) even with a residual term for gradient estimation bias correction (GEP).

## 5.3 OBSERVATIONS AND HIGHLIGHTS

**Better Solving the Long Tail Problem.** Learning from private data aims to extract unique knowledge, particularly from hard "long-tail" samples, i.e. samples with a large initial loss under the base model $m^{(0)}$, that are often most valuable for downstream tasks. While DP constraints prevent perfect learning, the goal is still to maximize information extraction from those samples within the privacy budget. To evaluate this, we track per-sample loss and accuracy, focusing on these long-tail samples.

In Fig. 3, we compare PE-SGD with the standard baseline DP-SGD on private evaluation set $\mathcal{A}$. Fig. 3(a) shows per-sample loss and accuracy differences (PE-SGD minus DP-SGD) after training, demonstrating that PE-SGD yields lower loss (negative values) and higher accuracy (positive values), with stronger improvements on samples with a high initial loss under $m^{(0)}$. This indicates that

`PE-SGD` better mitigates the long-tail problem, resulting in broader knowledge coverage. Fig. 3(b) further confirms this with a leftward shift in per-sample loss distribution.

In addition, Table 2 shows average loss reduction on the $10\%$ hardest samples with the largest initial loss under $m^{(0)}$. `PE-SGD` outperforms all methods, and its variants surpass DP-SGD, underscoring the importance of using the whole gradient subspace for projection.

**Better Per-step Update.** Overall, `PE-SGD` outperforms DP-SGD. To understand the source of this advantage, we further compare their behaviors at the level of single training steps by performing single-step updates with DP-SGD and `PE-SGD` along the same vanilla SGD fine-tuning trajectory, and evaluating the updated models using $\mathcal{A}$. Results are shown in Fig. 4, which demonstrate that, within each updation step, `PE-SGD` decreases loss and increases accuracy to a larger extent compared to DP-SGD, even when the model is near convergence at the later steps. See Appendix D.9 for more results.

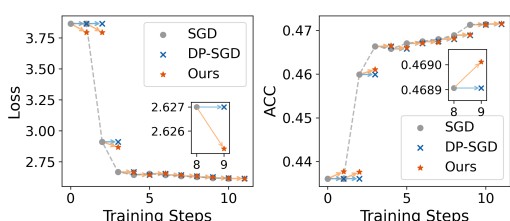

Figure 4: Per-step loss and accuracy (ACC) using different training methods along the same SGD trajectory on Qwen2.5 for Congressional Speech.

## 5.4 ABLATIONS

**Different Pre-trained LLMs.** Aside from Qwen2.5, we repeat our experiments on Llama3.2 and GPT2 to testify the universality of our method across different models. We can see from Fig. 5, `PE-SGD` still outperforms all baselines when using different models, with significantly lower loss and higher accuracy. More results are included in Appendix D.2.

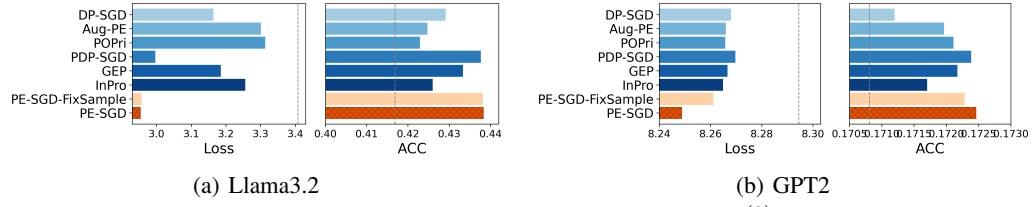

(a) Llama3.2
(b) GPT2

Figure 5: Loss and accuracy (ACC) comparison with different models $m^{(0)}$ for Congressional Speech.

**Different Amount of Private Sample ($M$).** We also perform experiments with larger private training dataset $\mathcal{B}$ to study the scalability and application range of `PE-SGD`. Results are included in Fig. 6. Note that smaller $\beta = 0.05$ and larger $T = 40$ are applied with larger $M \geq 1,000$. Results demonstrate that, with an increase in $M$, the performance of `PE-SGD` continues to improve, demon-

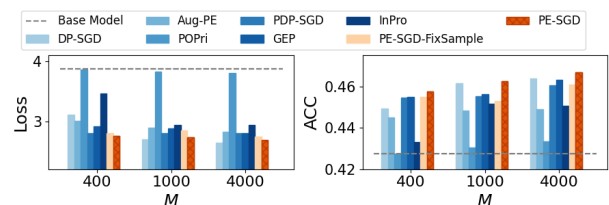

Figure 6: Loss and accuracy (ACC) comparison with different number of private training samples ($M$). Qwen2.5 is used for Congressional Speech.

strated by a decreasing loss and an increasing accuracy. Although the performance of DP-SGD increases dramatically with the increase of $M$, making it a very strong baseline, `PE-SGD` still performs better than or similar to it. Other methods all exhibit a performance increase with the increase in $M$.

**Different Differential Privacy Budget $\epsilon$.** In Fig. 7, we compare the performance of the trained model $m$ using `PE-SGD` and the most important baseline DP-SGD under $\epsilon = 1.0, 2.0, 4.0, 8.0$. The results show that, regardless of the DP setting, `PE-SGD` consistently outperforms DP-SGD, achieving lower loss and higher accuracy. Results for more baseline methods are included in Appendix D.3.

**Different Noise Addition Place.** In Algorithm 1, we add Gaussian noise to the final projection coefficient $\sum_{j=0}^{\tilde{M}}[(\boldsymbol{G}^T\boldsymbol{G})^{-1}(\boldsymbol{G}^T\boldsymbol{H})]_{:,j}$ to guarantee $(\epsilon, \delta)$-DP. Like mentioned in *[S2]* in Section 4.2, noise can also be added to the private gradient $\sum_{j=0}^{\tilde{M}}\boldsymbol{H}_{:,j}$ ("PE-SGD-NoisyRealGrad") and the

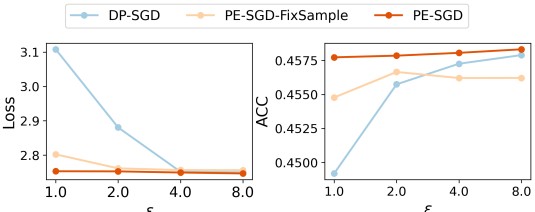
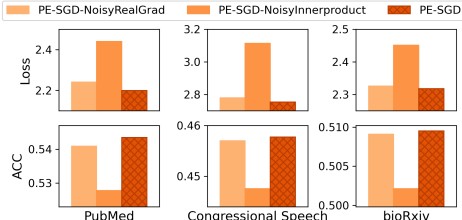

Figure 7: Comparison of trained model performance under different DP budge $\epsilon$ with Qwen2.5 and Congressional Speech.

Figure 8: Comparison of trained model performance under different noise addition place with Qwen2.5.

gradient inner product $\sum_{j=0}^{\tilde{M}}[\boldsymbol{G}^T\boldsymbol{H}]_{:,j}$ ("PE-SGD-NoisyInnerproduct"). Results in Fig. 8 shows that,"PE-SGD-NoisyInnerproduct" performs apparently worse than its counterparts, while adding noise to the final gradient coefficient (PE-SGD) outperforms adding noise to $\sum_{j=1}^{\tilde{M}}\boldsymbol{H}_{:,j}$, achieving a lower loss and a slightly higher accuracy. Since normalization (or clipping) is applied before aggregation, one key determinant of final performance is how close the column norms are to each other. If the norms are identical to one anther, no information is lost during respective normalization. Empirically, in Table 5 in Appendix D.4, we compare the STD-to-Mean and Min-to-Max ratios of column norms of the matrix to which noise is added to in these methods and find that PE-SGD achieves the lowest STD-to-Mean ratio and highest Min-to-Max ratio, confirming smaller norm disparities. In contrast, "PE-SGD-NoisyInnerproduct" exhibits extremely high STD-to-Mean and low Min-to-Max ratios, indicating much larger disparities.

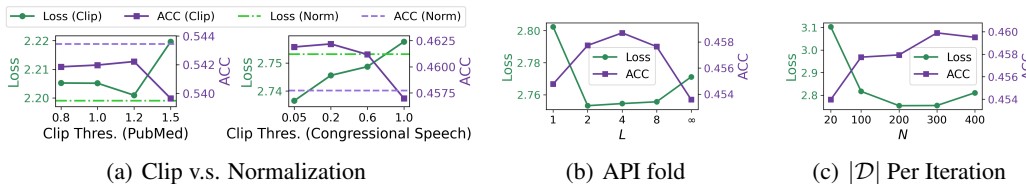

(a) Clip v.s. Normalization     (b) API fold     (c) $|\mathcal{D}|$ Per Iteration

Figure 9: Comparison of trained model performance: $(a)$ using normalization or different clipping threshold; $(b)$ under different generative API fold ($L$ in Algorithm 1); $(c)$ under different number of synthetic data per iteration $N$. Qwen2.5 is used for Congressional Speech (unless otherwise notated).

**Normalization and Clipping.** Both methods can be used to bound the $\ell_2$ sensitivity in DP algorithms. However, like shown in Algorithm 1 line 8, we choose to simply normalize each column of the gradient coefficient $\boldsymbol{Z}_{:,j} = [(\boldsymbol{G}^T\boldsymbol{G})^{-1}(\boldsymbol{G}^T\boldsymbol{H})]_{:,j}$ (i.e. $\boldsymbol{Z}_{:,j} = \boldsymbol{Z}_{:,j}/||\boldsymbol{Z}_{:,j}||_2$) instead of clipping (i.e. $\boldsymbol{Z}_{:,j} = C\boldsymbol{Z}_{:,j}/\max(||\boldsymbol{Z}_{:,j}||_2, C)$) with an additional hyper-parameter $C$ as clipping threshold. See Fig. 9(a) for a comparison of these two methods. Though one may achieve slightly better results using clipping, it requires careful tuning of the clipping threshold $C$. Since normalization avoids the need of tuning an additional hyper-parameter, we chose normalization for simplicity.

**Different Generative API fold $L$.** Another hyper-parameter in our study is $L$, the fold of VARIATIONAL_API(). Specifically, each selected seed sample in $\mathcal{D}^{seed}$ generates $L-1$ variations through the VARIATIONAL_API() of the updated model to evolve $\mathcal{D}$ (see Algorithm 3 for details). In our experiments, we use mainly $L = 2$. Notably, "PE-SGD-FixSample" corresponds to $L = 1$, where all samples in $\mathcal{D}$ are chosen as $\mathcal{D}^{seed}$ and no new variants are generated. In contrast, setting $L = \infty$ results in $|\mathcal{D}^{seed}| = 0$ and $\mathcal{D}$ being fully regenerated at each iteration using only the RANDOM_API() of the updated model. As shown in Fig. 9(b), when fixing $N = |\mathcal{D}| = 200$ per iteration, both extremes ($L = 1$ or $L = \infty$) degrade performance, whereas intermediate choices yield only minor performance differences.

**Different Number of Synthetic Data Per Iteration ($N$).** We further study the impact of $N$ on PE-SGD. Results are included in Fig. 9(c) showing that PE-SGD is able to gain good performance with only a few hundred samples (around 200 of samples). Therefore, PE-SGD is very sample efficient. This also verifies that the gradients of training are limited to a low-dimension space, much smaller than the number of training parameters $p$ (Li et al., 2022).

---

**Algorithm 2** PE-SGD with Data Synthesis using More Powerful Generative Models

---

**Input:** Pre-trained generative model $m^{(0)}$ with parameter $\phi$; More power generative model or API $u$ with un-trainable or freezing parameters; empty synthetic dataset $\mathcal{D} \leftarrow \emptyset$; private dataset $\mathcal{B}$ of size $M$; number of iterations for updates $T$; number of synthetic samples to generate $N$ per iteration; number of synthetic sample variation fold $L$; Poisson sub-sampling ratio $\beta$ for private samples; learning rate $\eta$; DP parameters $\epsilon, \delta$;
**Output:** Fine-tuned or trained PLM $m$.

1: $\mathcal{D}_{seed}^{(0)} \leftarrow \emptyset, \mathcal{D} \leftarrow \texttt{SynDataGeneration}(\ \mathcal{D}_{seed}^{(0)}, N, L, u\ )$.
2: $m \leftarrow$ fine-tuned $m^{(0)}$ using $\mathcal{D}$.
3: $\sigma \leftarrow \texttt{NoiseCalculation}(\ \epsilon, \delta, T, \beta\ )$ following Gopi et al. (2021).
4: **for** $t = 1$ **to** $T$ **do**
5: $\quad \mathcal{B}^{(t)} \leftarrow \texttt{PoissonSubSampling}(\ \mathcal{B}, \beta\ ), \tilde{M} \leftarrow |\mathcal{B}^{(t)}|$.
6: $\quad \boldsymbol{G} \leftarrow [\boldsymbol{g}_1, \ldots, \boldsymbol{g}_N]$ with $\boldsymbol{g}_i$ being the sample-level gradient of $\mathbf{x}_i \in \mathcal{D}$ on $m$.
7: $\quad \boldsymbol{H} \leftarrow [\boldsymbol{h}_1, \ldots, \boldsymbol{h}_{\tilde{M}}]$ with $\boldsymbol{h}_j$ being the sample-level gradient of $\mathbf{w}_j \in \mathcal{B}^{(t)}$ on $m$.
8: $\quad \boldsymbol{z} \leftarrow \texttt{SumUpNormalizedColumns}(\ (\boldsymbol{G}^T\boldsymbol{G})^{-1}(\boldsymbol{G}^T\boldsymbol{H})\ ) + \mathcal{N}(0, \sigma^2 I_N)$.
9: $\quad \phi \leftarrow \phi + \eta \cdot [(\boldsymbol{G}\boldsymbol{z})/\tilde{M}]$ to update $m$.
10: $\quad \mathcal{D}_{seed}^{(t)} \leftarrow \texttt{SampleSelection}(\ \boldsymbol{z}, N, L, \mathcal{D}\ ), \mathcal{D} \leftarrow \texttt{SynDataGeneration}(\ \mathcal{D}_{seed}^{(t)}, N, L, u\ )$.
11: **end for**

---

**Synthesizing Using More Powerful Generative Models.** To further improve performance, PE-SGD can be easily extended to "generating synthetic samples using a stronger model or API, then performing DP fine-tuning on a weaker model using these synthetic samples" by freezing the parameters of the stronger generative model while fine-tuning the weaker model (see Algorithm 2 for detail). This extension helps to exploit a more powerful model for synthesizing data of higher quality. In Table 3, we compare two settings: (1) generating synthetic data using the fine-tuned Qwen2.5-3B-Instruct itself, and (2) generating using a more powerful model, namely Qwen2.5-7B-Instruct or GPT-4o-mini, while still fine-tuning the same Qwen2.5-3B-Instruct target model. Results clearly show that using the stronger generation model leads to consistent performance gains over relying on Qwen2.5-3B-Instruct itself alone.

Table 3: Fine-tuning Qwen2.5-3B-Instruct using synthetic samples generated by different generative models. $M = 400, \beta = 0.2, N = 200, T = 10$ is applied.

| Generation Model | $\epsilon$ | PubMed | | Congressional Speech | | bioRxiv | |
|---|---|---|---|---|---|---|---|
| | | Loss ($\downarrow$) | ACC ($\uparrow$) | Loss ($\downarrow$) | ACC ($\uparrow$) | Loss ($\downarrow$) | ACC ($\uparrow$) |
| Qwen2.5-3B-Instruct | | $2.1990_{\pm0.0124}$ | $0.5435_{\pm0.0009}$ | $2.7532_{\pm0.0166}$ | $0.4577_{\pm0.0011}$ | $2.3178_{\pm0.0018}$ | $0.5095_{\pm0.0000}$ |
| Qwen2.5-7B-Instruct | 1.0 | $2.1879_{\pm0.0319}$ | $\mathbf{0.5480}_{\pm0.0025}$ | $\mathbf{2.7152}_{\pm0.0233}$ | $0.4615_{\pm0.0022}$ | $2.3042_{\pm0.0023}$ | $0.5139_{\pm0.0003}$ |
| GPT-4o-mini | | $\mathbf{2.1862}_{\pm0.0051}$ | $\mathbf{0.5480}_{\pm0.0007}$ | $2.7219_{\pm0.0367}$ | $\mathbf{0.4618}_{\pm0.0013}$ | $\mathbf{2.3013}_{\pm0.0031}$ | $\mathbf{0.5142}_{\pm0.0002}$ |
| Qwen2.5-3B-Instruct | | $2.1839_{\pm0.0131}$ | $0.5436_{\pm0.0005}$ | $2.7468_{\pm0.0291}$ | $0.4579_{\pm0.0021}$ | $2.3165_{\pm0.0040}$ | $0.5097_{\pm0.0004}$ |
| Qwen2.5-7B-Instruct | $\infty$ | $2.1794_{\pm0.0340}$ | $0.5493_{\pm0.0026}$ | $\mathbf{2.7104}_{\pm0.0266}$ | $0.4620_{\pm0.0019}$ | $2.3015_{\pm0.0030}$ | $\mathbf{0.5148}_{\pm0.0004}$ |
| GPT-4o-mini | | $\mathbf{2.1737}_{\pm0.0089}$ | $\mathbf{0.5495}_{\pm0.0007}$ | $2.7198_{\pm0.0359}$ | $\mathbf{0.4622}_{\pm0.0013}$ | $\mathbf{2.3010}_{\pm0.0024}$ | $\mathbf{0.5148}_{\pm0.0002}$ |

Due to space limitation, more experiments are included in Appendix D considering synthetic sample SFT enhanced DP-SGD, different number of training iteration $T$, efficiency of PE-SGD and so on.

## 6 CONCLUSION AND FUTURE WORK

In this work, we propose PE-SGD, a gradient projection based evolutionary framework for differentially private training. Extensive experiments demonstrate that PE-SGD surpasses classical DP-SGD in training high-performance models especially under tight DP budgets with limited private data. In particular, PE-SGD shows strong effectiveness in addressing the long-tail problem. Its sample efficiency makes it a practical solution for real-world applications.

Future work includes extending verification to other modalities, exploring training from scratch, and refining evolutionary strategies to better control the $\ell_2$ distance between DP-protected approximate gradients and real private gradients (Fig. 1).

REPRODUCIBILITY STATEMENT

For reproducibility, we provide the following materials:

- Source code that is applied as supplementary material during submission and open-sourced at `https://github.com/LindaLydia/PE-SGD`.
- Detailed experimental settings included in Section 5.1 and Appendix C including private data curation, synthetic data generation prompt and learning hyper-parameters.

ACKNOWLEDGMENTS

Author Tianyuan Zou and author Yang Liu acknowledge the support of the Presidential Young Scholar Scheme (Project No. P0056638), RIFL (Project No. 4-CG00), the RIAIoT (Project No. P0059914) at The Hong Kong Polytechnic University.

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

APPENDIX

## A LARGE LANGUAGE MODEL USAGE STATEMENT

Large Language Models, particularly the GPT series and Gemini, have been applied to API usage demonstrations, code generation for plotting, auto-completion (with GitHub Copilot), academic paper writing refinement, and grammar correction.

## B ADDITIONAL INFORMATION FOR ALGORITHM AND METHODS

### B.1 DETAIL ALGORITHM FUNCTIONS

In this section, we elaborate the functions used in Algorithm 1 in detail in Algorithm 3.

---

**Algorithm 3** Functions used in Algorithm 1 for `PE-SGD`

---

**function** `SynDataGeneration`( $\mathcal{D}^{seed}$, $N$, $L$, $m$ ):
    **if** $L == \infty$ **then**
        $\mathcal{D}^{seed} \leftarrow N$ samples generated from $m$ using `RANDOM_API()`.
        $\mathcal{D}^{var} \leftarrow \emptyset$
    **else**
        **if** $|\mathcal{D}^{seed}| == 0$ **then**
            $\mathcal{D}^{seed} \leftarrow \lceil N/L \rceil$ samples generated from $m$ using `RANDOM_API()`.
        **end if**
        $\mathcal{D}^{var} \leftarrow \emptyset$.
        **for** $\mathbf{x}_i \in \mathcal{D}^{seed}$ **do**
            $\mathcal{D}^{var} \leftarrow \mathcal{D}^{var} \bigcup \{ L - 1$ samples generated from $m$ using `VARIATIONAL_API`($\mathbf{x}_i$)$\}$.
        **end for**
    **end if**
    **return** $\mathcal{D}^{seed} \bigcup \mathcal{D}^{var}$.

**function** `PoissonSubSampling`( $\mathcal{B}$, $\beta$ ):
    $\hat{\mathcal{B}} \leftarrow \emptyset$.
    **for** $\mathbf{w}_j \in \mathcal{B}$ **do**
        $X \sim$ Bernoulli($\beta$).
        **if** X==1 **then**
            $\hat{\mathcal{B}} \leftarrow \hat{\mathcal{B}} \bigcup \{\mathbf{w}_j\}$.
        **end if**
    **end for**
    **return** $\hat{\mathcal{B}}$.

**function** `SumUpNormalizedColumns`( $\boldsymbol{Z}$ ):
    Denote $\boldsymbol{Z} = (z_{i,j})_{1 \leq i \leq N; \, 1 \leq j \leq \tilde{M}}$ where $\boldsymbol{Z} \in \mathbb{R}^{N \times \tilde{M}}$ and $\boldsymbol{Z}_{:,j} = [z_{1,j}, \ldots, z_{N,j}]^T$.
    **for** $j = 1$ **to** $\tilde{M}$ **do**
        **for** $i = 1$ **to** $N$ **do**
            $z_{i,j} \leftarrow z_{i,j}/||\boldsymbol{Z}_{:,j}||_2$, where $||\boldsymbol{Z}_{:,j}||_2 \leftarrow \sqrt{\sum_{i=1}^{N} z_{i,j}^2}$ being the $\ell_2$ norm of $\boldsymbol{Z}_{:,j}$.
        **end for**
    **end for**
    **return** $\sum_{j=1}^{\tilde{M}} \boldsymbol{Z}_{:,j}$.

**function** `SampleSelection`( $\boldsymbol{z}$, $N$, $L$, $\mathcal{D}$ ):
    Denote $\boldsymbol{z} = [z_i, \ldots, z_N]^T$ with $\boldsymbol{z} \in \mathbb{R}^N$.
    $\boldsymbol{z} \leftarrow |\boldsymbol{z}|$, i.e. $\boldsymbol{z} \leftarrow [|z_i|, \ldots, |z_N|]^T$.
    $\boldsymbol{p}_z = \frac{\exp(\boldsymbol{z})}{\sum_{i=1}^{N} \exp(z_i)}$.
    Random sample $\lceil N/L \rceil$ different index $\mathcal{S} = \{s_1, \ldots, s_{\lceil N/L \rceil}\}$ according to the probability given by $\boldsymbol{p}_z$.
    **return** $\{\mathbf{x}_i \, | \, \mathbf{x}_i \in \mathcal{D}, i \in \mathcal{S}\}$.

---

## B.2 PRIVACY ANALYSIS FOR PE-SGD

In this section, we prove that Algorithm 1 guarantees $(\epsilon, \delta)$-DP.

**Theorem B.1.** PE-SGD *(Algorithm 1) has the exact same $(\epsilon, \delta)$-DP guarantees as usual DP-SGD for the same noise scale $\sigma$, sub-sampling probability $\beta$ and number of iterations $T$ (i.e., the composition of sub-sampled Gaussian mechanism).*

*Proof.* In Algorithm 1, within each iteration, the only point where private information, namely the private sample gradients $\boldsymbol{H}$, is exposed is in the computation $\boldsymbol{Z} = (\boldsymbol{G}^T \boldsymbol{G})^{-1}(\boldsymbol{G}^T \boldsymbol{H})$ (line 8). In this place, the $i^{th}$ column of $Z$ only depends on the $i^{th}$ column of $\boldsymbol{H}$ and thus only depends on the $i^{th}$ private sample. By the post-processing property of differential privacy (Dwork et al., 2014), any further computation on the output of an $(\hat{\epsilon}, \hat{\delta})$-DP mechanism does not incur additional privacy loss. Hence, as long as DP is guaranteed for the aggregated quantity $\boldsymbol{z} = \sum_{j=1} \boldsymbol{Z}_{:,j}$ in line 8 in Algorithm 1, DP is guaranteed for the whole iteration. To protect this sensitive result $\boldsymbol{z}$, we therefore normalize each column of $Z$ to have $\ell_2$ norm 1 and add isotropic Gaussian noise $\sigma \times \mathcal{N}(0, I)$ to it.

Thus, the problem reduces to the composition of sub-sampled Gaussian mechanism in exactly the same way and with the same parameters as DP-SGD. This question has been carefully studied in several prior works such as Abadi et al. (2016). In particular, we use the PRV Accountant[4] from Gopi et al. (2021) which computes the exact privacy budget to arbitrary accuracy. PRV Accountant has also been adopted as the default privacy accountant in Opacus,[5] a widely used PyTorch library for differentially private training, which is regarded as the standard DP framework in both academia and industry. □

## B.3 THE RELATIONSHIP BETWEEN DIFFERENT PROJECTION METHODS

In this section, we discuss the relationship between the projection method given in PDP-SGD (Zhou et al., 2021), GEP (Yu et al., 2021) and our projection method given in Eq. (3).

**Theorem B.2.** *If full eigenspace $\boldsymbol{F} \in \mathbb{R}^{p \times r}$ instead of the top-$k$ ($k < r$) eigenspace $\boldsymbol{E} \in \mathbb{R}^{p \times k}$ of $\boldsymbol{G}\boldsymbol{G}^T$ is exploited in PDP-SGD (Zhou et al., 2021), their projection function $\hat{\boldsymbol{G}} = \boldsymbol{F}\boldsymbol{F}^T \boldsymbol{H}$ is equal to our projection method $\hat{\boldsymbol{G}} = \boldsymbol{G}(\boldsymbol{G}^T \boldsymbol{G})^{-1}(\boldsymbol{G}^T \boldsymbol{H})$ given in Eq. (3).*

*Proof.* First of all, from linear algebra, $rank(\boldsymbol{G}) = rank(\boldsymbol{G}\boldsymbol{G}^T) = r$. Therefore, the singular vector decomposition (SVD) of $\boldsymbol{G}$ can be written as $\boldsymbol{G} = \boldsymbol{U}\boldsymbol{\Sigma}\boldsymbol{V}^T$ where $\boldsymbol{U} \in \mathbb{R}^{p \times r}$ and $\boldsymbol{V} \in \mathbb{R}^{N \times r}$ have orthonormal columns ($\boldsymbol{U}^T \boldsymbol{U} = \boldsymbol{V}^T \boldsymbol{V} = \boldsymbol{I}_r \in \mathbb{R}^{r \times r}$) and $\boldsymbol{\Sigma} = \text{diag}(\sigma_1, \ldots, \sigma_r)$ with $\sigma_i > 0, i = 1, \ldots, r$.

Then $\boldsymbol{G}\boldsymbol{G}^T = (\boldsymbol{U}\boldsymbol{\Sigma}\boldsymbol{V}^T)(\boldsymbol{V}\boldsymbol{\Sigma}\boldsymbol{U}^T) = \boldsymbol{U}\boldsymbol{\Sigma}^2\boldsymbol{U}^T$. Therefore, the eigenvectors of $\boldsymbol{G}\boldsymbol{G}^T$ are the columns of $\boldsymbol{U}$. If $\boldsymbol{F}$ spans the eigenspace of $\boldsymbol{G}\boldsymbol{G}^T$ corresponding to the non-zero eigenvalues, then we have $\boldsymbol{F} = \boldsymbol{U}$ with $\boldsymbol{F}\boldsymbol{F}^T = \boldsymbol{U}\boldsymbol{U}^T$.

Note that $\boldsymbol{G}^T \boldsymbol{G} = \boldsymbol{V}\boldsymbol{\Sigma}\boldsymbol{U}^T \boldsymbol{U}\boldsymbol{\Sigma}\boldsymbol{V}^T = \boldsymbol{V}\boldsymbol{\Sigma}^2 \boldsymbol{V}^T$, thus $(\boldsymbol{G}^T \boldsymbol{G})^{-1} = \boldsymbol{V}\boldsymbol{\Sigma}^{-2}\boldsymbol{V}^T$.

Therefore,

$$\begin{aligned}
\boldsymbol{G}(\boldsymbol{G}^T \boldsymbol{G})^{-1}\boldsymbol{G}^T &= (\boldsymbol{U}\boldsymbol{\Sigma}\boldsymbol{V}^T)(\boldsymbol{V}\boldsymbol{\Sigma}^{-2}\boldsymbol{V}^T)(\boldsymbol{V}\boldsymbol{\Sigma}\boldsymbol{U}^T) \\
&= \boldsymbol{U}\boldsymbol{\Sigma}\boldsymbol{I}_r\boldsymbol{\Sigma}^{-2}\boldsymbol{I}_r\boldsymbol{\Sigma}\boldsymbol{U}^T \\
&= \boldsymbol{U}\boldsymbol{I}_r\boldsymbol{U}^T = \boldsymbol{U}\boldsymbol{U}^T = \boldsymbol{F}\boldsymbol{F}^T.
\end{aligned}$$

Multiplying by $\boldsymbol{H}$ at the end, we get $\boldsymbol{G}(\boldsymbol{G}^T \boldsymbol{G})^{-1}(\boldsymbol{G}^T \boldsymbol{H}) = \boldsymbol{F}\boldsymbol{F}^T \boldsymbol{H}$. □

**Theorem B.3.** *If all left singular vectors $\boldsymbol{F} \in \mathbb{R}^{p \times r}$ of $\boldsymbol{G}$ instead of $\boldsymbol{E} \in \mathbb{R}^{p \times k}$, the top-$k$ ($k < r$) left singular vectors of $\boldsymbol{G}$ corresponds to the top-$k$ singular values ($k < r$), is exploited in GEP (Yu et al., 2021), their projection function $\hat{\boldsymbol{G}} = \boldsymbol{F}\boldsymbol{F}^T \boldsymbol{H}$ (without considering the residual term $\boldsymbol{R}$) is equal to our projection method $\hat{\boldsymbol{G}} = \boldsymbol{G}(\boldsymbol{G}^T \boldsymbol{G})^{-1}(\boldsymbol{G}^T \boldsymbol{H})$ given in Eq. (3).*

---

[4]An open-source implementation that is available at `https://github.com/microsoft/prv_accountant`.
[5]Opacus Homepage: `https://opacus.ai`.

*Proof.* Similar as above, the singular vector decomposition (SVD) of $G$ can be written as $G = U\Sigma V^T$ where $U \in \mathbb{R}^{p \times r}$ and $V \in \mathbb{R}^{N \times r}$ have orthonormal columns ($U^T U = V^T V = I_r \in \mathbb{R}^{r \times r}$) and $\Sigma = \mathrm{diag}(\sigma_1, \ldots, \sigma_r)$ with $\sigma_i > 0, i = 1, \ldots, r$. If $F$ spans the whole left singular vectors space of $G$, then we have $F = U$ with $FF^T = UU^T$.

Thus, when neglecting the residual term of GEP, the projection of GEP is the same as that of PDP-SGD. From Theorem B.2, the projection method of GEP is also equal to $\hat{G} = G(G^T G)^{-1}(G^T H)$ given in Eq. (3).

$\square$

## C  IMPLEMENTATION DETAIL

### C.1  DATASET CURATION

- PubMed (NLM (U.S.), 2025). As samples later than January $8^{th}$ 2025 are not published, we use samples that has complete date between December $1^{st}$ 2024 and January $8^{th}$ 2025. We also remove samples that are less than 100 characters in length. We randomly partition the remaining samples with train:dev:test=6:1:3 resulting in a total of 8613 training samples, 1435 dev samples and 4307 test samples.

- Congressional Speech (Hou et al., 2025). We use samples from January 2025 to May 2025 as the training part and samples from June 2025 as the test part. We further performs a random partition on the training part with train:dev=4:1. We then removes "None" samples and samples less than 5 characters, resulting in a total of 33413 training samples, 8229 dev samples and 19880 test samples.

- bioRxiv (Hou et al., 2025). We also use samples from January 2025 to May 2025 as the training part and samples from June 2025 as the test part. We then randomly partitioned the training part with train:dev=6:1. This results in a total of 13775 training samples, 2296 dev samples and 2507 test samples.

In our experiments, $\mathcal{B}$ is randomly selected from the training partition while $\mathcal{A}$ uses the whole test partition or randomly selected $4,000$ samples from the test partition for Congressional Speech as the dataset is too large. To align with other methods, we pass the original text input directly to tokenizer, instead of using "encode_plus()" after "tokenize()" as originally done by POPri for GPT2 model.

### C.2  API PROMPT

In Table 4, we listed the prompts used in our experiments, including zero-shot in-context learning prompts for RANDOM_API and one-shot prompt for VARIATIONAL_API(). We mainly follow Xie et al. (2024) for the construction of these prompts. The zero-shot prompt is also used for evaluation. Note that we use chat template throughout the training and evaluation process.

### C.3  LEARNING HYPER-PARAMETERS

In our experiments, we use LoRA fine-tuning with LoRA rank $r = 8$, LoRA $\alpha = 32$ and LoRA dropout rate $0.1$. For Qwen2.5 and Llama3.2, we perform LoRA on "q_proj" and "v_proj" modules, whereas for GPT2 we perform LoRA on "c_attn" and "c_proj" modules.

The learning rate $\eta$ varies across tasks. With $M = 400$, $\eta = 5e^{-4}$ when training Qwen2.5 for PubMed and Congressional Speech; $\eta = 3e^{-4}$ when training Qwen2.5 for bioRxiv; $\eta = 2e^{-4}$ when training Llama3.2 for Congressional Speech; $\eta = 3e^{-4}$ when training Llama3.2 for PubMed; $\eta = 1e^{-5}$ when training GPT2 for all tasks. With $M = 1,000$ and $M = 4,000$, $\eta = 4e^{-4}$ and $\eta = 3e^{-4}$ are used respectively for training Qwen2.5 for Congressional Speech. Warmup ratio is set to $0.08$ and weight decay ratio set to $0.01$. AdamW is used as the optimizer. Each experiment is repeated using 3 random seeds.

If clipping is applied, unless otherwise stated, the clipping threshold is set to $C = 1.0$, except the clipping threshold for the residual term in GEP (Yu et al., 2021), for which the clipping threshold is set to 0.2 following the clipping threshold given in their paper (Yu et al., 2021).

Table 4: Prompt used for synthetic dataset generation. Due to clarity, we omit the words in the parentheses in the labels of Openreview-Category and the attributes of Openreview-Rating.

| Dataset | Prompt Type | Prompt | <tone> |
|---|---|---|---|
| PubMed | zero-shot | "Using a variety of sentence structures, write an abstract (only the abstract, without other parts) for a medical research paper: " | - |
| | one-shot | "*<sample>*\nBased on the above sample, using a variety of sentence structures *<tone>*, write an abstract (only the abstract, without other parts) for medical research paper: " | "in a professional way", "in a professional tone", "in a professional style", "in a concise manner", "in a creative style", "using imagination", "in a storytelling tone", "in a formal manner" |
| Congressional Speech | zero-shot | "Using a variety of oral sentence structures, write a line from a public debating (beginning with the appropriate formal address for the chamber) or speech delivered in the United States Congress, the UK Parliament, or the Canadian Parliament: " | - |
| | one-shot | "*<sample>*\nBased on the above sample, using a variety of oral sentence structures *<tone>*, write a line from a public debating (beginning with the appropriate formal address for the chamber) or speech delivered in the United States Congress, the UK Parliament, or the Canadian Parliament: " | "in a occasionally adversarial but civil manner", "with a strong sense of patriotism", "in a formal and official tone", "showing deep respect and decorum", "using a respectful and measured style", "in a persuasive and compelling way", "adopting an argumentative tone", "with creative and imaginative language", "employing rhetorical flourishes", "in a thoughtful and reasoned manner", "with passionate conviction", "using clear and direct speech", "in a diplomatic and tactful style", "expressing urgency and determination", "with nuanced and balanced arguments" |
| bioRxiv | zero-shot | "Using a variety of sentence structures, write a realistic bioRxiv preprint abstract (only the abstract, without other parts) for a life science (e.g. biology, neuroscience, cognition, etc.) research paper: " | - |
| | one-shot | "*<sample>*\nBased on the above sample, using a variety of sentence structures *<tone>*, write a realistic bioRxiv preprint abstract (only the abstract, without other parts) for a life science (e.g. biology, neuroscience, cognition, etc.) research paper: " | "in a professional way", "in a professional tone", "in a professional style", "in a concise manner", "in a creative style", "using imagination", "in a storytelling tone", "in a formal manner", "in a high-quality manner", "in a low-quality manner" |

## D ADDITIONAL RESULTS

### D.1 SIMILARITY BETWEEN APPROXIMATE DP-PROTECTED GRADIENTS AND REAL PRIVATE GRADIENTS

In Fig. 10, we additionally show the cosine similarity between approximate DP-protected gradients with real private gradients under different ablation methods by removing one of the two key components used in `PE-SGD`: empirically better noise placement, and dynamic projection subspace that evolves with training. This figure testifies the importance of combining all these two components to get good results.

Moreover, to better understand why a fixed projection subspace leads to suboptimal performance, specifically, the growing $\ell_2$ discrepancy

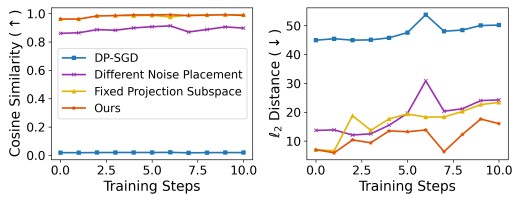

Figure 10: Similarity between approximate DP-protected (with noise) gradient and private gradient (without noise). See Section 1 for detail.

between the projected and true private gradients as training progresses (see Fig. 10), we analyze the principal angles (Gu et al., 2025) between the private gradient subspace at the initial iteration, $\boldsymbol{H}^{(0)}$, and those at later iterations, $\boldsymbol{H}^{(t)}$ for $t = 1, \ldots, T-1$. To compute these angles, we first extract the top-50 orthonormal basis vectors of each subspace, yielding $\boldsymbol{E}^{(0)}, \boldsymbol{E}^{(t)} \in \mathbb{R}^{p \times 50}$. We then

form $\boldsymbol{O} = \boldsymbol{E}^{(0)^\top} \boldsymbol{E}^{(t)}$ and compute its singular values $s_j$ for $j = 1, \ldots, 50$ via SVD. The principal angles follow as $\theta_j = \arccos(s_j)$, lying in $[0°, 90°]$. Larger angles indicate greater divergence between subspaces, with $90°$ corresponding to orthogonality. As shown in Fig. 11, these principal angles increase steadily over the progression of training for all three tasks (private datasets) and different fine-tune models. This consistent increase demonstrates that the private gradient subspace evolves significantly during fine-tuning, independent of the specific dataset or fine-tuning model. These findings underscores our claim that relying on a fixed projection subspace is suboptimal for fine-tuning.

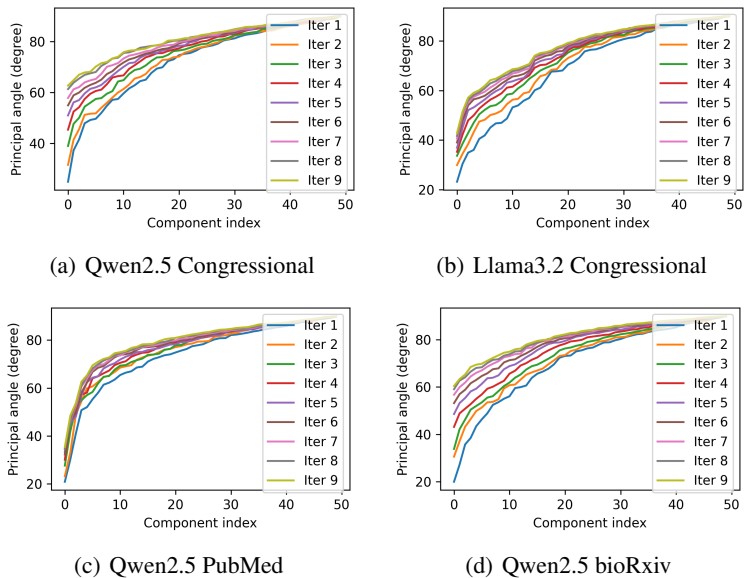

(a) Qwen2.5 Congressional        (b) Llama3.2 Congressional

(c) Qwen2.5 PubMed        (d) Qwen2.5 bioRxiv

Figure 11: Principle angles of initial private gradient subspace and private gradient subspaces in later fine-tuning iteration. $M = 400, \beta = 0.2, L = 2, T = 10, N = 200$ is applied.

## D.2 DIFFERENT PRE-TRAINED LLMS

In this section, we include more results for Llama3.2 and GPT2 in Fig. 12. Other settings remain the same as that for main experiments, i.e. $M = 400, \beta = 0.2, T = 10, \delta = 1e^{-5}, N = 200, L = 2$. PE-SGD demonstrates consistent superiority compared to other methods.

## D.3 DIFFERENT DP BUDGET $\epsilon$

In this section, we examine how the performance of the fine-tuned model $m$ varies with the differential privacy budget $\epsilon$ across all baseline methods. Results are included in Fig. 13. While the curves, particularly those for methods other than DP-SGD, appear nearly flat, their values do change slightly with $\epsilon$. However, these variations are minor compared to the substantial performance gaps between methods. This suggests that the relative performance differences are primarily method-driven. Moreover, the robustness of most gradient projection methods to DP noise (high performance under tight budge scenario) underscores the effectiveness of reducing noise dimensionality through projection.

## D.4 COLUMN NORM DISPARITY OF DIFFERENT MATRIX

We include the numerical results of STD-to-Mean ratio and Min-to-Max ratio of column norms of $\boldsymbol{H}$, $\boldsymbol{G}^T \boldsymbol{H}$, $(\boldsymbol{G}^T \boldsymbol{G})^{-1}(\boldsymbol{G}^T \boldsymbol{H})$ respectively for the three different noise addition strategies, namely "PE-SGD-NoisyRealGrad", "PE-SGD-NoisyInnerproduct" and PE-SGD in Table 5. Results shows smaller norm disparities of PE-SGD compared to others, with "PE-SGD-NoisyInnerproduct" exhibiting much larger disparities.

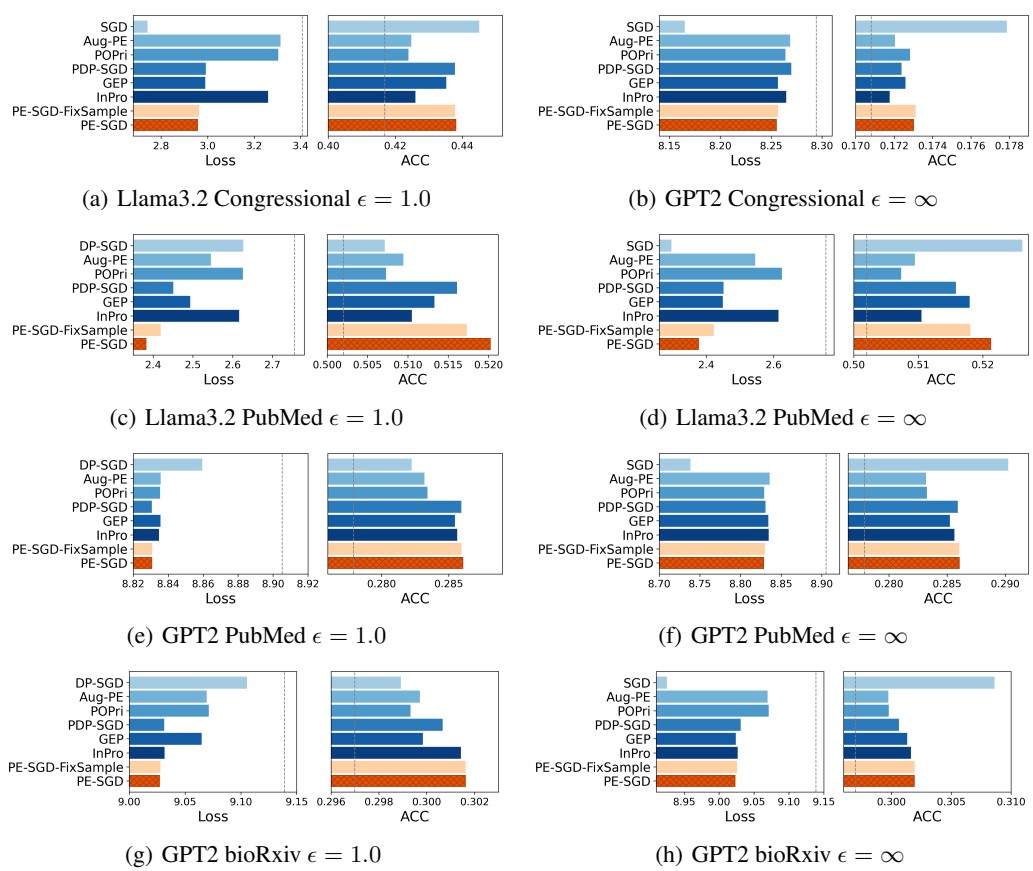

Figure 12: Loss and accuracy (ACC) comparison with different models $m^{(0)}$.

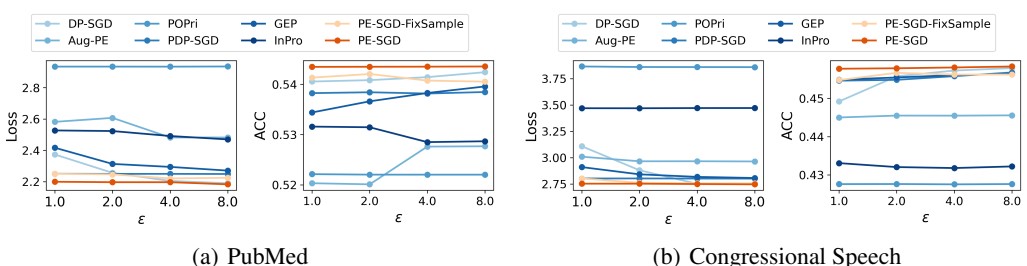

Figure 13: Comparison of loss and accuracy (ACC) of fine-tuned model $m$ under different DP budge $\epsilon$ using Qwen2.5.

Table 5: Comparison of the STD-to-Mean ratio and Min-to-Max ratio of column norms in the noise-attached-to matrix across three different noise addition strategies. Results are averaged across iterations first and then across three random seeds. **Best** results are marked.

| | PE-SGD-NoisyRealGrad | PE-SGD-NoisyInnerproduct | PE-SGD |
|---|---|---|---|
| Norms | $\left\{\|\boldsymbol{H}_{:,j}\|_2\right\}_{j=1}^{\tilde{M}}$ | $\left\{\left\|[\boldsymbol{G}^T\boldsymbol{H}]_{:,j}\right\|_2\right\}_{j=1}^{\tilde{M}}$ | $\left\{\left\|[(\boldsymbol{G}^T\boldsymbol{G})^{-1}(\boldsymbol{G}^T\boldsymbol{H})]_{:,j}\right\|_2\right\}_{j=1}^{\tilde{M}}$ |
| STD / Mean ($\downarrow$) | $0.443_{\pm 0.020}$ | $0.533_{\pm 0.015}$ | $\mathbf{0.404}_{\pm 0.009}$ |
| Min / Max ($\uparrow$) | $0.094_{\pm 0.007}$ | $0.039_{\pm 0.002}$ | $\mathbf{0.121}_{\pm 0.006}$ |

### D.5 Impact of Generation Prompt

To assess the impact of different prompt templates on data synthesis, we follow prior work (Xie et al., 2024) and report final model performance as well as FID, Precision, and Recall of the synthetic dataset in Table 6 on Congressional Speech task using Qwen2.5. Note that the FID, Precision and Recall metrics are computed using 200 randomly generated samples from the fine-tuned model after the whole training process. The first row corresponds to the prompt used in our main results ("Original"), which was designed following prior work (Xie et al., 2024). "P-1" removes all tone options from the one-shot prompt while keeping the zero-shot prompt unchanged. "P-2" keeps only three tones in the one-shot prompt, again with the zero-shot prompt unchanged relative to "Original". "P-3" and "P-4" use prompts rewritten by ChatGPT, with "P-3" being more structured and "P-4" being more creativity-oriented (as judged by ChatGPT). Full prompt details are provided in Table 7.

Table 6: Comparison of final model performance, FID, Precision, Recall of the generated synthetic dataset using different prompts throughout the training process of PE-SGD using Qwen2.5.

| Prompt | Loss ($\downarrow$) | ACC ($\uparrow$) | FID ($\downarrow$) | Precision ($\uparrow$) | Recall ($\uparrow$) |
|---|---|---|---|---|---|
| Original | $2.7532_{\pm 0.0166}$ | $\mathbf{0.4577}_{\pm 0.0011}$ | $0.1571_{\pm 0.0056}$ | $\mathbf{0.9200}_{\pm 0.0537}$ | $0.5235_{\pm 0.1032}$ |
| P-1 | $2.7526_{\pm 0.0136}$ | $0.4570_{\pm 0.0012}$ | $0.1625_{\pm 0.0121}$ | $0.8533_{\pm 0.0153}$ | $0.5416_{\pm 0.1860}$ |
| P-2 | $\mathbf{2.7522}_{\pm 0.0336}$ | $0.4572_{\pm 0.0012}$ | $\mathbf{0.1524}_{\pm 0.0048}$ | $0.8633_{\pm 0.0681}$ | $0.5233_{\pm 0.1015}$ |
| P-3 | $2.7915_{\pm 0.0493}$ | $0.4576_{\pm 0.0025}$ | $0.1767_{\pm 0.0135}$ | $0.7200_{\pm 0.1609}$ | $\mathbf{0.6483}_{\pm 0.1703}$ |
| P-4 | $2.8017_{\pm 0.0995}$ | $0.4574_{\pm 0.0020}$ | $0.2054_{\pm 0.0097}$ | $0.6933_{\pm 0.1320}$ | $0.5708_{\pm 0.3621}$ |

From Table 6 we observe that using different reasonable prompts has only a minor impact on final model performance (Loss and ACC). However, different prompts affect the Precision and Recall of the synthetic dataset with respect to the private dataset.

On one hand, although less diversity inspiring prompts ("P-1" and "P-2" v.s. "Original") preserve final model performance and FID, they significantly reduce Precision, indicating an increasing number of diverging samples that are less correlated to the target private distribution. On the other hand, a more structured prompt ("P-3" v.s. "Original") improves Recall (covering more of the private sample space) but harms Precision (introducing more synthetic samples that deviate from the private distribution); while a more creative-inspiring prompt ("P-4") performs worse overall than the others.

### D.6 Visualization of Synthetic and Real Gradient Subspace

We include a PCA visualization of the synthetic and real gradient subspaces of PE-SGD in Figs. 14(a), 14(c) and 14(e). These figures show that their alignment improves substantially over the course of training compared with the initial iteration, and is better than that of the fixed-projection baseline ("PE-SGD-FixSample" in Figs. 14(b), 14(d) and 14(f)). This provides clear evidence for the effectiveness of the evolving projection mechanism in PE-SGD.

### D.7 The Enhancement of Synthetic Data SFT

As shown in Algorithm 1 (line 13), our pre-trained model fine-tuning procedure builds on a synthetic data supervised fine-tuning (SFT) of $m^{(0)}$. As illustrated in Table 8, this step provides additional performance gains for $m$, although not to a large extent.

### D.8 SFT Enhanced DP-SGD

Like shown in Algorithm 1, our pre-trained model fine-tuning is based on a synthetic data supervised fine-tuning (SFT) of $m^{(0)}$ with helps to improve the performance of the final model $m$ like shown in Table 8. This would lead to a straightforward improved baseline for DP-SGD, i.e. first SFT on synthetic data and then performs DP-SGD on the tuned model. We compare PE-SGD with this improved baseline ("SFT + DP-SGD") in Table 9. We also display the results of synthetic data SFT (the first step of "SFT + DP-SGD") to give detailed information of this 2 stage procedure. Results demonstrate that, even with this SFT enhancement, PE-SGD still outperforms the SFT improved

Table 7: Different prompts used for synthetic dataset generation in Table 6.

| Dataset | Prompt Type | Prompt | \<tone\> |
|---|---|---|---|
| P-1 for Congressional Speech | zero-shot | "Using a variety of oral sentence structures, write a line from a public debating (beginning with the appropriate formal address for the chamber) or speech delivered in the United States Congress, the UK Parliament, or the Canadian Parliament: " | - |
| | one-shot | "\<sample\>\nBased on the above sample, using a variety of oral sentence structures, write a line from a public debating (beginning with the appropriate formal address for the chamber) or speech delivered in the United States Congress, the UK Parliament, or the Canadian Parliament: " | - |
| P-2 for Congressional Speech | zero-shot | "Using a variety of oral sentence structures, write a line from a public debating (beginning with the appropriate formal address for the chamber) or speech delivered in the United States Congress, the UK Parliament, or the Canadian Parliament: " | - |
| | one-shot | "\<sample\>\nBased on the above sample, using a variety of oral sentence structures \<tone\>, write a line from a public debating (beginning with the appropriate formal address for the chamber) or speech delivered in the United States Congress, the UK Parliament, or the Canadian Parliament: " | "in a formal and official tone", "in a persuasive and compelling way", "using clear and direct speech" |
| P-3 for Congressional Speech | zero-shot | "Compose one line of dialogue suitable for a formal public debate or parliamentary speech. The line should begin with the proper formal address for the chamber and reflect the speaking style of either the United States Congress, the UK Parliament, or the Canadian Parliament. " | - |
| | one-shot | "\<sample\>\nUsing the sample above as reference, and adopting \<tone\> expression, compose one line of dialogue suitable for a formal public debate or parliamentary speech. The line should begin with the proper formal address for the chamber and reflect the speaking style of either the United States Congress, the UK Parliament, or the Canadian Parliament." | "in a occasionally adversarial but civil manner", "with a strong sense of patriotism", "in a formal and official tone", "showing deep respect and decorum", "using a respectful and measured style", "in a persuasive and compelling way", "adopting an argumentative tone", "with creative and imaginative language", "employing rhetorical flourishes", "in a thoughtful and reasoned manner", "with passionate conviction", "using clear and direct speech", "in a diplomatic and tactful style", "expressing urgency and determination", "with nuanced and balanced arguments" |
| P-4 for Congressional Speech | zero-shot | "Craft a single line as if spoken in a public debate or parliamentary speech. Begin with an appropriate formal salutation for the floor, and write it in the style of a speech from the U.S. Congress, the British Parliament, or the Canadian Parliament. " | - |
| | one-shot | "\<sample\>\nDrawing inspiration from the given sample and using \<tone\> phrasing, craft a single line as if spoken in a public debate or parliamentary speech. Begin with an appropriate formal salutation for the floor, and write it in the style of a speech from the U.S. Congress, the British Parliament, or the Canadian Parliament." | "in a occasionally adversarial but civil manner", "with a strong sense of patriotism", "in a formal and official tone", "showing deep respect and decorum", "using a respectful and measured style", "in a persuasive and compelling way", "adopting an argumentative tone", "with creative and imaginative language", "employing rhetorical flourishes", "in a thoughtful and reasoned manner", "with passionate conviction", "using clear and direct speech", "in a diplomatic and tactful style", "expressing urgency and determination", "with nuanced and balanced arguments" |

DP-SGD ("SFT + DP-SGD"), achieving lower loss and higher accuracy when DP is guaranteed ($\epsilon = 1.0$). Also, it's not surprising that under $\epsilon = \infty$ (without DP protection), "SFT + DP-SGD"

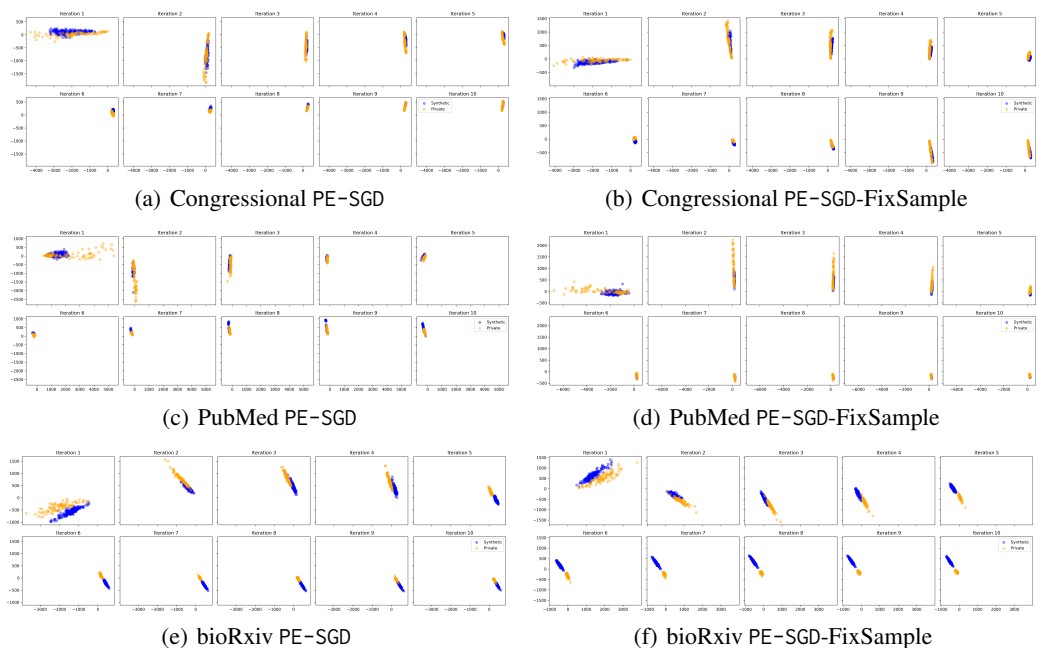

(a) Congressional PE-SGD

(b) Congressional PE-SGD-FixSample

(c) PubMed PE-SGD

(d) PubMed PE-SGD-FixSample

(e) bioRxiv PE-SGD

(f) bioRxiv PE-SGD-FixSample

Figure 14: Visualization of synthetic and private gradient subspaces within each fine-tuning iteration using PCA. $M = 400, \beta = 0.2, L = 2, T = 10, N = 200$ is applied using Qwen2.5. Blue points represent synthetic gradients while orange samples represent private gradients.

Table 8: Comparison of PE-SGD with and without (w/o) synthetic data SFT procedure using Qwen2.5. **Better** results are marked.

| Method | $\epsilon = 1.0$ | | $\epsilon = \infty$ | |
|---|---|---|---|---|
| | Loss ($\downarrow$) | Accuracy ($\uparrow$) | Loss ($\downarrow$) | Accuracy ($\uparrow$) |
| PE-SGD w/o SFT | $2.7740_{\pm 0.0183}$ | $0.4559_{\pm 0.0006}$ | $2.7471_{\pm 0.0007}$ | $0.4578_{\pm 0.0008}$ |
| PE-SGD | $\mathbf{2.7532}_{\pm 0.0166}$ | $\mathbf{0.4577}_{\pm 0.0011}$ | $\mathbf{2.7468}_{\pm 0.0095}$ | $\mathbf{0.4579}_{\pm 0.0021}$ |

performs better. This is because, it now trains directly on real private sample gradients that is only clipped but not perturbed, whereas PE-SGD still trains on a combination of synthetic sample gradients.

Table 9: Comparison of PE-SGD and synthetic data SFT enhanced DP-SGD ("SFT + DP-SGD") using Qwen2.5. **Better** results between "SFT + DP-SGD" and PE-SGD are marked.

| Method | PubMed | | Congressional Speech | | bioRxiv | |
|---|---|---|---|---|---|---|
| | $\varepsilon = 1$ | $\varepsilon = \infty$ | $\varepsilon = 1$ | $\varepsilon = \infty$ | $\varepsilon = 1$ | $\varepsilon = \infty$ |
| **Next Token Prediction Loss ($\downarrow$)** | | | | | | |
| Base Model | 2.9341 | | 3.8690 | | 2.5311 | |
| SGD | $2.1197_{\pm 0.0004}$ | | $2.6078_{\pm 0.0084}$ | | $2.2756_{\pm 0.0001}$ | |
| SFT | $2.5174_{\pm 0.0477}$ | | $3.3767_{\pm 0.2430}$ | | $2.5261_{\pm 0.0052}$ | |
| SFT + DP-SGD | $2.2579_{\pm 0.0107}$ | $\mathbf{2.1621}_{\pm 0.0061}$ | $2.8821_{\pm 0.0337}$ | $\mathbf{2.7193}_{\pm 0.0329}$ | $2.3773_{\pm 0.0067}$ | $\mathbf{2.2891}_{\pm 0.0013}$ |
| PE-SGD | $\mathbf{2.1990}_{\pm 0.0124}$ | $2.1839_{\pm 0.0131}$ | $\mathbf{2.7532}_{\pm 0.0166}$ | $2.7468_{\pm 0.0095}$ | $\mathbf{2.3178}_{\pm 0.0018}$ | $2.3165_{\pm 0.0040}$ |
| **Next Token Prediction Accuracy (ACC) ($\uparrow$)** | | | | | | |
| Base Model | 0.5217 | | 0.4276 | | 0.4981 | |
| SGD | $0.5494_{\pm 0.0004}$ | | $0.4661_{\pm 0.0013}$ | | $0.5145_{\pm 0.0004}$ | |
| SFT | $0.5269_{\pm 0.0032}$ | | $0.4364_{\pm 0.0054}$ | | $0.4982_{\pm 0.0009}$ | |
| SFT + DP-SGD | $0.5419_{\pm 0.0004}$ | $\mathbf{0.5450}_{\pm 0.0006}$ | $0.4537_{\pm 0.0013}$ | $\mathbf{0.4673}_{\pm 0.0019}$ | $0.5075_{\pm 0.0003}$ | $\mathbf{0.5126}_{\pm 0.0001}$ |
| PE-SGD | $\mathbf{0.5435}_{\pm 0.0009}$ | $0.5436_{\pm 0.0005}$ | $\mathbf{0.4577}_{\pm 0.0011}$ | $0.4579_{\pm 0.0021}$ | $\mathbf{0.5095}_{\pm 0.0000}$ | $0.5097_{\pm 0.0004}$ |

## D.9 PER-STEP UPDATE

In this section, we include more results on other datasets to demonstrate the effectiveness of `PE-SGD` compared to baseline DP-SGD within each single update. Like shown in Fig. 15, results on 2 other datasets, PubMed and bioRxiv, all demonstrate the effectiveness of `PE-SGD` in single step training update, resulting in lower loss and higher accuracy.

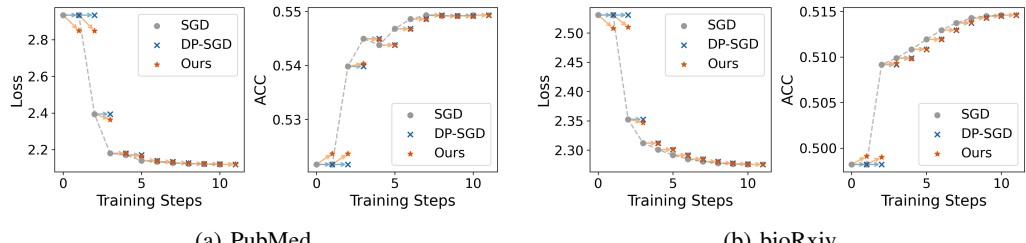

(a) PubMed             (b) bioRxiv

Figure 15: Per-step loss and accuracy (ACC) using different training methods along the same SGD trajectory on Qwen2.5.

## D.10 DIFFERENT NUMBER OF TRAINING ITERATIONS $T$

In this section, we examine the effect of the number of iterations (or parameter update steps) $T$ has on the final performance by comparing `PE-SGD` with the most important baseline DP-SGD. Results are included in Fig. 16. Note that, when the same $(\epsilon, \delta)$-DP is guaranteed, $T$ also affects the noise level $\sigma$ for each iteration. We can see from Fig. 16 that `PE-SGD` consistently outperforms DP-SGD regardless of $T$. Moreover, using longer training iterations (larger $T$) incurs just minor performance updates, especially for `PE-SGD`, while the performance increase for DP-SGD is relatively larger when $T \leq 20$. Therefore, the we conclude that, `PE-SGD` is a more effective training method, given its ability in achieving higher performance within fewer updates steps.

We further clarify that training with $T = 10$, each using on average $\mathbb{E}[\tilde{M}] = M \times \beta = 80$ samples, suffices for convergence given the limited private data ($M = 400$). As shown in Figs. 4 and 15, the SGD trajectory converges within 10 updates (training steps), with the evaluation loss remaining nearly unchanged.

## D.11 DIFFERENT PROMPTING RESULT OF POPRI

For the POPri baseline (Hou et al., 2025), we strictly follow their training prompts during DPO training. During the evaluation of the Qwen2.5 and Llama3.2 models fine-tuned with instruction, instruction is also used (i.e., instruction-following evaluation), with loss and accuracy computed only on tokens from evaluation samples. Since different prompts yield different performance, we report results using the better-performing prompt (lower loss, higher accuracy) in our main paper. Our prompt (see Table 4 for detail) consistently outperforms theirs, so we use it for main results (Table 1), and report evaluation results using their prompts in Table 10 as well.

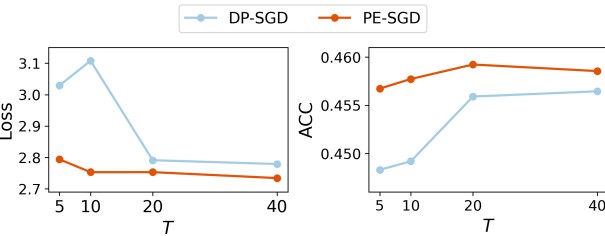

Figure 16: Loss and accuracy (ACC) under different number of training iterations (update steps). Qwen2.5 is used for Congressional Speech.

Table 10: Evaluation comparison of the same fine-tuned model from Qwen2.5 using POPri but with different prompts.

| Prompt Source | Method | | PubMed | | Congressional Speech | | bioRxiv | |
|---|---|---|---|---|---|---|---|---|
| | | | Loss ($\downarrow$) | Accuracy ($\uparrow$) | Loss ($\downarrow$) | Accuracy ($\uparrow$) | Loss ($\downarrow$) | Accuracy ($\uparrow$) |
| POPri | Base Model | | 2.9836 | 0.5207 | 4.0727 | 0.4256 | 2.7413 | 0.4828 |
| | POPri $\epsilon = 1.0$ | | 2.9612 | 0.5221 | 3.9846 | 0.4271 | 2.6948 | 0.4861 |
| | $\epsilon = \infty$ | | 2.9549 | 0.5178 | 3.9922 | 0.4258 | 2.7225 | 0.4813 |
| Our | Base Model | | 2.9341 | 0.5217 | 3.8690 | 0.4276 | 2.5311 | 0.4981 |
| | POPri $\epsilon = 1.0$ | | 2.9344 | 0.5221 | 3.8656 | 0.4276 | 2.6178 | 0.4930 |
| | $\epsilon = \infty$ | | 2.9487 | 0.5192 | 3.8700 | 0.4276 | 2.5309 | 0.4981 |

## D.12 TRAINING EFFICIENCY OF PE-SGD

In this section, we compare the per-iteration runtime of exploiting one sub-sampled private batch from $\mathcal{B}$ (i.e., a full DPO training step for POPri and a single parameter update for other methods) across PE-SGD, DP-SGD, and baselines. We use the default setting $M = 400, \beta = 0.2, N = 200, L = 2$ and fine-tune Qwen2.5 for Congressional Speech. For DP-SGD, we adopt the Opacus package with its built-in gradient accumulation to ensure exactly one model update per sub-sampled batch. Results are shown in Table 11.

Unsurprisingly, gradient projection based methods (PDP-SGD, GEP, PE-SGD) incur higher computational costs than DP-SGD, as they compute synthetic gradients $G$ in addition to private gradients $H$, with $G \in \mathbb{R}^{p \times N}$ being significantly larger than $H \in \mathbb{R}^{p \times \tilde{M}}$ ($\mathbb{E}[\tilde{M}] = 80$ v.s. $N = 200$). Compared to PE-SGD, PDP-SGD and GEP reduce cost modestly by retaining only the top-$k$ eigenvalues ($k = 50$ v.s. $N = 200$). POPri is by far the slowest, as each iteration requires several DPO update steps. GEP costs a little bit longer than PDP-SGD as it calculates the residual term.

There might be concerns that generating additional synthetic samples for synthetic dataset (gradient subspace) evolution is in-efficient. In fact, although evolving the synthetic dataset across iterations does introduce additional computational overhead, the required amount of synthetic data per iteration is small: around 200 samples per iteration when using generative API fold $L = 2$ (i.e., only around 100 newly generated samples each iteration), as shown in Fig. 9(c). Consequently, the added cost remains modest and manageable in practice. This is further supported by the empirical generation times for 100 synthetic samples using Qwen2.5-3B-Instruct, reported in Table 12.

Table 11: Computational efficiency comparison between different methods. Time are measured for one training iteration, the exploitation of one sub-sampling batch from private dataset $\mathcal{B}$ (i.e. entire DPO training for POPri and one-time parameter update for others), without considering the time for synthetic data generation.

| | | DP-SGD | POPri | PDP-SGD | GEP | PE-SGD |
|---|---|---|---|---|---|---|
| Time [s] | | 9.7 | 743.8 | 127.3 | 135.2 | 148.6 |

Table 12: Time costs (seconds, s) for generating 100 synthetic samples with Qwen2.5-3B-Instruct model using different generation APIs on three tasks. Tested with 80G-A100.

| | RANDOM_API | VARIATIONAL_API |
|---|---|---|
| PubMed | 26 | 41 |
| Congressional Speech | 24 | 36 |
| bioRxiv | 42 | 62 |

## D.13 DISCUSSION ON FIG. 3(A)

One interesting phenomenon in Fig. 3(a) is that some data points fall precisely on the diagonal line $(y = x)$, as well as on several other straight lines. These patterns correspond to subsets of examples with specific token lengths in the real private dataset of Congressional Speech dataset. This dataset consists of truncated public speech segments. Some samples contained therein are very short, for example, "The hon. Member." or "I declare the motion carried.". This indicates that there are only a limited number of possible combinations for how many tokens are predicted correctly or incorrectly before and after training. As a result, these short samples naturally cluster on lines like $y = x$, where the $x$-axis represents accuracy before training (base-model accuracy) and the $y$-axis represents the difference in accuracy after training using PE-SGD compared to using DP-SGD. For example, the second sample contains 7 tokens, of which only 1 token is correctly predicted by the base model, while 2 and 4 tokens are correctly predicted after fine-tuning using DP-SGD and PE-SGD respectively, resulting in a point with $x = y = \frac{2}{7}$.

Table 13: "Overlap" metric of the synthetic and private gradient subspace at each iteration. Qwen2.5-3B-Instruct is used with $M = 400, \beta = 0.2, T = 10, N = 200$. Higher values indicate higher similarity between subspaces.

| Iteration | PubMed | | Congressional Speech | | bioRxiv | |
|---|---|---|---|---|---|---|
| | w/ evolution | w/o evolution | w/ evolution | w/o evolution | w/ evolution | w/o evolution |
| 1 | 11.87 | 11.87 | 6.69 | 6.69 | 7.11 | 7.11 |
| 2 | 11.64 | 9.88 | 5.80 | 5.59 | 5.56 | 5.58 |
| 3 | 11.93 | 9.41 | 4.88 | 5.52 | 5.39 | 5.29 |
| 4 | 6.36 | 8.89 | 5.06 | 4.97 | 5.77 | 4.59 |
| 5 | 8.72 | 7.64 | 4.78 | 4.56 | 3.75 | 3.46 |
| 6 | 8.10 | 7.54 | 5.74 | 4.14 | 3.71 | 3.10 |
| 7 | 6.54 | 6.37 | 5.38 | 3.98 | 3.13 | 2.91 |
| 8 | 6.88 | 6.51 | 4.71 | 4.51 | 3.40 | 3.18 |
| 9 | 7.81 | 6.30 | 5.67 | 5.00 | 3.68 | 3.51 |
| 10 | 8.57 | 6.85 | 6.16 | 5.27 | 3.89 | 3.68 |

# E CONVERGENCE COMPARISON OF PE-SGD WITH PREVIOUS WORKS

We provide a convergence analysis below by demonstrating that "synthetic sample (gradient subspace) evolution improves the convergence of differentially private gradient projection algorithms with high probability".

First, we need to clarify that, the key point of this proof is substantially different from prior works on DP gradient projection (e.g., PDP-SGD (Hou et al., 2025) and GEP (Yu et al., 2021)). Those works implicitly assume that all synthetic sample gradients follow the same distribution as the private gradients. However, as empirically demonstrated in Table 13 (the "overlap" metric between synthetic and private gradient subspaces[6] is far smaller than 50, which indicates that the two subspaces are identical) this assumption does not hold in practice. In contrast, the goal of PE-SGD is precisely to push the gradient subspace of synthetic samples to align with that of the private samples. Therefore, our proof focuses on characterizing the gradient alignment of evolution.

More specifically, Theorem 3.3 in GEP (Yu et al., 2021) shows that the convergence rate is positively correlated with the average squared norm of the gradient-projection residual term. Thus, the central goal of our proof is to show that evolution reduces the upper bound of this average squared residual term with high probability, which directly improves the convergence behavior under the DP constraints.

## E.1 NOTATIONS AND ASSUMPTIONS

We assume that $N$ is a multiple of $L$, and we have $\hat{N} = N/L$ synthetic samples, each with corresponding gradient $\boldsymbol{g}_i \in \mathbb{R}^{p \times 1}$. Following Algorithm 1, after using VARIATIONAL_API for next

---

[6]The "overlap" metric (Zhang et al., 2025) is defined as the "overlap" between $\boldsymbol{G}^{(t)} \in \mathbf{R}^{p \times N}$ and $\boldsymbol{H}^{(t)} \in \mathbb{R}^{p \times \tilde{M}}$. Here, we use $overlap = \sum_{j=1}^{50} s_j^2$, with $s_{j=1,\dots,50}$ being the singular values of $\boldsymbol{O} = {\boldsymbol{E}_G^{(t)}}^T \boldsymbol{E}_H^{(t)}$ and $\boldsymbol{E}_G^{(t)}, \boldsymbol{E}_H^{(t)}$ being the top-50 orthonormal basis of $\boldsymbol{G}^{(t)}, \boldsymbol{H}^{(t)}$ respectively (Gu et al., 2025).

generation synthetic dataset creation, we get another $\hat{N}(L-1)$ samples, with corresponding gradients $\{\boldsymbol{g}_i\}_{i=\hat{N}+1}^{N}$. Let $\boldsymbol{h} \in \mathbb{R}^{p \times 1}$ be the averaged gradient from the private samples.

We make the following assumptions:

- We use the clipping method to bound the sensitivity (see corresponding discussion and experiments in Section 5.4), and assume that the $\ell_2$ norm of the gradient projection coefficients from each private sample are below the clipping threshold. In this way, we can focus our analysis on the impact of the added DP noise.

- We assume that the synthetic gradients have $\ell_2$ norm 1 (i.e., $\|\boldsymbol{g}_i\| = 1, \forall i = 1, \cdots, N$) and are orthogonal to each other (i.e., $< \boldsymbol{g}_i, \boldsymbol{g}_j >= 0, \forall i \neq j$).

- Let $z_i = (\boldsymbol{g}_i^\top \boldsymbol{g}_i)^{-1} \boldsymbol{g}_i^\top \boldsymbol{h} = \frac{\boldsymbol{g}_i^\top \boldsymbol{h}}{\boldsymbol{g}_i^\top \boldsymbol{g}_i} = \frac{\boldsymbol{g}_i^\top \boldsymbol{h}}{\|\boldsymbol{g}_i\|^2} = \boldsymbol{g}_i^\top \boldsymbol{h}$ be the clean gradient projection coefficient, and $|z_i|$ is lower bounded by $\zeta > 0$. We assume the variant sample $x_{i'}$ of $x_i$ created using the VARIATIONAL_API satisfies $z_{i'} = z_i + \zeta$ with probability $\frac{1}{2}$ and $z_{i'} = z_i - \zeta$ with probability $\frac{1}{2}$. This represents the setting where the VARIATIONAL_API could either improve or hurt the projection with the same probability.

- To select the samples for the next iteration, we add i.i.d. DP noise $\nu \sim \mathcal{N}(0, \sigma^2)$ to the projection coefficient $z_i$, and select the samples with the top noisy coefficient from each variation group. We assume that the index of the selected samples are $s_1, \ldots, s_{\hat{N}}$.

## E.2 THEOREM AND DISCUSSION

**Theorem E.1.** *(High-Probability Improvement from Gaussian-Perturbed Variants) With a large possibility ($> 1 - \tau$), the selected $\hat{N}$ samples have a sum of squared projection coefficients not smaller than that of the original $\hat{N}$ samples:*

$$\Pr(\sum_{i=1}^{\hat{N}} z_{s_i}^2 \geq \sum_{i=1}^{\hat{N}} z_i^2) \geq 1 - \tau,$$

*where $\tau = \frac{N}{L} \cdot \exp\left(-\frac{(L-1)}{2} \cdot \Phi(\frac{\zeta}{2\sigma})\right)$, and $\Phi$ is the CDP of $\mathcal{N}(0,1)$.*

***Discussion:*** This theorem (result), Theorem E.1, directly implies that the norm of the projected gradients $\left\|\sum_{i=1}^{\hat{N}} z_i \boldsymbol{g}_i\right\| = \sqrt{\sum_{i=1}^{\hat{N}} z_i^2}$ increases with high probability across iterations. Consequently, the norm of the residual gradients on the selected $\hat{N}$ samples, an upper bound on the residual gradients across all $N$ samples, decreases with a high probability over iterations.

## E.3 PROOF

To prove Theorem E.1, we first prove Lemma E.2 as follows.

**Lemma E.2.** *(VARIATIONAL_API Improvement on Each Group) Let $x_i$ be a synthetic sample, with $L-1$ samples $\{x_{i,l}\}_{l=1}^{L-1}$ generated from the VAIRATIONAL_API using $x_i$ as the seed sample, i.e. $x_{i,l}$ is a variant of $x_i$. Denote $x_i$ as $x_{i,0}$ for clarity. Let $z_{i,l}$ be the gradient projection coefficient of $x_{i,l}$, and $v_{i,l} = z_{i,l} + n_{i,l}$ be the coefficient with DP noise where $n_{i,l} \sim \mathcal{N}(0, \sigma^2)$. Let $k_i = \arg\max_{0 \leq l \leq L-1} v_{i,l}$ be the index of the selected sample. We have that with high probability, the selected sample improves the projection coefficient:*

$$\Pr(|z_{i,k_i}| \geq |z_{i,0}|) \geq 1 - \exp\left(-\frac{(L-1)}{2} \cdot \Phi(\frac{\zeta}{2\sigma})\right).$$

We first prove Lemma E.2.

*Proof.* Without loss of generality, we assume that $z_{i,0} > 0$. For variants $j$ such as $z_{i,j} = z_{i,0} + \zeta$, we have:

$$\Pr(v_{i,j} < v_{i,0}) \leq 1 - \Phi(\frac{\zeta}{2\sigma}),$$

where $\Phi$ is the CDP of $\mathcal{N}(0, 1)$.

As the $L-1$ variants generated using the VARIATIONAL_API are independent, we have:

$$\Pr(|z_{i,k_i}| \leq |z_{i,0}|) \leq \Pr\left(\forall 1 \leq l < L : (z_{i,l} = z_{i,0} - \zeta) \text{ or } (z_{i,l} = z_{i,0} + \zeta \text{ and } v_{i,l} < v_{i,0})\right)$$

$$= \left(\frac{1}{2} + \frac{1}{2}(1 - \Phi(\frac{\zeta}{2\sigma}))\right)^{L-1} = \left(1 - \frac{1}{2} \cdot \Phi(\frac{\zeta}{2\sigma})\right)^{L-1} \leq \exp\left(-\frac{(L-1)}{2} \cdot \Phi(\frac{\zeta}{2\sigma})\right).$$

The last inequality holds as $1 - x \leq e^{-x}$ (after using first order Talyer expansion). Therefore,

$$\Pr(|z_{i,k_i}| \geq |z_{i,0}|) \geq 1 - \exp\left(-\frac{(L-1)}{2} \cdot \Phi(\frac{\zeta}{2\sigma})\right).$$

$\square$

With Lemma E.2, we prove our main theorem Theorem E.1.

*Proof.* Since each group of variations is operated independently, we have

$$\Pr(\sum_{i=1}^{\hat{N}} z_{s_i}^2 \geq \sum_{i=1}^{\hat{N}} z_i^2) \geq \Pr(\forall 1 \leq i \leq \hat{N} : |z_{i,k_i}| \geq |z_{i,0}|) = \left[1 - \exp\left(-\frac{(L-1)}{2} \cdot \Phi(\frac{\zeta}{2\sigma})\right)\right]^{N/L}$$

$$\geq 1 - \frac{N}{L} \cdot \exp\left(-\frac{(L-1)}{2} \cdot \Phi(\frac{\zeta}{2\sigma})\right).$$

$\square$

