# OpenReview forum: "PE-SGD: Differentially Private Deep Learning via Evolution of Gradient Subspace for Text"
_ICLR.cc/2026/Conference — ICLR 2026 Poster_

### Official Review · Reviewer_v9H2 · 2025-10-30

**Soundness:** 3
**Presentation:** 3
**Contribution:** 3
**Rating:** 6
**Confidence:** 4

**Summary:**

DP-SGD and its variants often suffer from degraded performance when the number of private samples is limited. Prior work attempts to address this issue by computing a gradient projection subspace from public data and adding perturbations within that subspace. However, there is a mismatch between the fixed projection subspace and the dynamically evolving training process, which hampers model performance. Moreover, the optimal location for injecting noise within the projection process remains underexplored.

This paper proposes PE-SGD, which dynamically updates the non-private dataset based on the evolving model during training, thereby reducing the L2 distance between the approximate gradients and the real gradients. In addition, the paper systematically compares different noise injection strategies and finds that adding noise to the final projection coefficients yields more stable and superior performance than injecting noise at other stages. Extensive experiments demonstrate the effectiveness of PE-SGD, particularly in improving performance on long-tailed samples within the private dataset.

**Strengths:**

- The paper identifies an important limitation in existing gradient projection approaches: the fixed projection subspace is incompatible with the dynamic nature of training. Figure 1 effectively illustrates how the subspace derived by prior methods diverges from the true gradient subspace as training progresses, revealing a key reason behind their performance degradation.
- Methodologically, the work integrates the strengths of gradient projection and differentially private synthetic data approaches (e.g., PDP-SGD and Aug-PE). It introduces a novel design that computes the projection subspace using dynamically generated samples, achieving lower next-token prediction loss and higher next-token prediction accuracy on text generation tasks.
- The paper provides comprehensive experiments to explain why PE-SGD outperforms prior methods. For example, analyses of long-tailed samples before and after training highlight that PE-SGD captures richer knowledge from private data and mitigates the long-tail issue. Moreover, per-step update experiments clearly demonstrate that PE-SGD achieves a larger decrease in loss compared to DP-SGD.

**Weaknesses:**

- In the SynDataGeneration component, the authors employ VARIATIONAL_API to generate synthetic data. However, the paper lacks an evaluation of the quality of these generated samples, such as their textual quality and diversity. This step is crucial for future extensions of the work, as insufficient diversity in data generated by VARIATIONAL_API could lead to performance degradation when applied to other datasets.

**Questions:**

- The reviewer would like to understand why, in the right plot of Figure 3(a), some data points appear to lie on the same diagonal line. Is this phenomenon due to numerical precision issues, or does it reflect certain regularities inherent in the data?

---

> ### Author Response · Authors · 2025-11-17
> **Response to Reviewer v9H2 (1/2)**
>
> We thank our reviewer for the time and effort dedicated to evaluating our work, as well as for the constructive comments and suggestions. Our detailed responses are provided below. We have also revised the paper to further improve clarity and have highlighted all changes in blue.
>
> **Response to [Weakness] Generation Quality Analysis:**
>
> We thank our reviewer for raising this interesting point. Overall, the quality of the generated synthetic dataset improves throughout training iterations.
>
> We report the quality of the generated synthetic datasets in Table (H) using FID, Precision, and Recall, standard metrics in DP synthetic data evaluation following prior work [1]. In Table (H), the “Training’’ rows correspond to the $200$ synthetic samples produced during PE-SGD training ($100$ from `RANDOM_API` or selected samples kept from previous iterations and $100$ from `VARIATIONAL_API`). The “Random’’ rows correspond to $200$ synthetic samples generated solely with `RANDOM_API`, using either the fine-tuned model or the base model prior to training. These “Random’’ results are included to directly reflect the fine-tuned model’s generation capability at various stages. Lower FID (closer alignment with the private dataset distribution), higher Precision (fewer diverging samples that are less correlated to the target private distribution), and higher Recall (wider coverage of the private dataset distribution) are deemed as better synthetic distribution indicators, demonstrating better quality and diversity.
>
> From Table (H), we observe a consistent decrease in FID and clear increases in both Precision and Recall after fine-tuning with PE-SGD, demonstrating the method’s effectiveness in improving synthetic data quality. Importantly, the benefits of synthetic dataset evolution are reaffirmed: the fixed synthetic dataset version of PE-SGD (PE-SGD-FixSample) achieves similar FID but significantly lower Precision and Recall after training compared to PE-SGD. This degradation explains the performance drop observed when applying PE-SGD-FixSample instead of the full PE-SGD procedure (see Table 1 in the paper).
>
> It is unsurprising that the “Random’’ rows consistently yield better embedding-based metrics than their “Training’’ counterparts.  This is because, during PE-SGD training, sample selection is **not** based on embedding similarity to the private dataset; rather, it is driven by the (absolute value of) gradient coefficient $z$ (see Algorithm 1, line 10). This encourages retaining samples that have gradients with stronger projections in the private gradient subspace. As a result, some selected samples may have large negative coefficients, reflecting useful gradient directions despite being relatively far from the private data distribution in embedding space. Consequently, embedding-distance metrics naturally favor “Random’’ samples, while PE-SGD training prioritizes samples that best support gradient projection.
>
> Table (H). Synthetic dataset quality and diversity assessment across training iterations using Qwen2.5-3B-Instruct for Congressional Speech with $M=400, \beta=0.2, T=10, N=200$. With FixSample, the same synthetic dataset is used across training iterations during training, therefore the values remain the same across iterations.
> | Method | Phase | FID | FID | FID | Precision | Precision | Precision | Recall | Recall | Recall |
> |:-:|:-:|:-:|:-:|:-:|:-:|:-:|:-:|:-:|:-:|:-:|
> |  |  | $iter=0$| $iter=5$ | $iter=10$ | $iter=0$| $iter=5$ | $iter=10$ | $iter=0$| $iter=5$ | $iter=10$ |
> | PE-SGD | Training | 0.2337 | 0.2095 | 0.1900 | 0.7767  | 0.8051 | 0.9300 | 0.0158  | 0.1575 | 0.3025  |
> | PE-SGD | Random | 0.2335 | 0.1763 | 0.1571 | 0.7953 | 0.8609 | 0.9200 | 0.0975 | 0.3296 | 0.5235 |
> | PE-SGD-FixSample | Training | 0.2337 | 0.2337 | 0.2337 | 0.7767 | 0.7767 | 0.7767 | 0.0158 | 0.0158 | 0.0158 |
> | PE-SGD-FixSample | Random | 0.2335 | 0.1853 | 0.1772 | 0.7953  | 0.7296 | 0.7800 | 0.0975 | 0.3751 | 0.4125 |
>
> ---
>
> [1] Xie, Chulin et al. "Differentially private synthetic data via foundation model apis 2: Text." ICML 2024.

---

> ### Author Response · Authors · 2025-11-17
> **Response to Reviewer v9H2 (2/2)**
>
> **Response to [Question] Lining points in Figure 3(a):**
>
> We thank our reviewer for your carefulness in reading and reviewing our paper, and we really appreciate your curiosity.
>
> We have examined this phenomenon in detail before the submission, and your observation is exactly correct. Some data points fall precisely on the diagonal line ($y = x$), as well as on several other straight lines. These patterns correspond to subsets of examples with specific token lengths in the real private dataset of the Congressional Speech dataset. This dataset consists of truncated public speech segments. Some samples contained therein are very short, for example, "The hon. Member." or “I declare the motion carried.”. This indicates that there are only a limited number of possible combinations for how many tokens are predicted correctly or incorrectly before and after training. As a result, these short samples naturally cluster on lines like $y = x$, where the $x$-axis represents accuracy before training (base-model accuracy) and the $y$-axis represents the difference in accuracy after training using PE-SGD compared to using DP-SGD. For example, the second sample contains 7 tokens, of which only 1 token is correctly predicted by the base model, while 2 and 4 tokens are correctly predicted after fine-tuning using DP-SGD and PE-SGD respectively, resulting in a point with $x=y=2/7$. We have added this discussion to the revised version of the paper in Appendix D.13.

---

### Official Review · Reviewer_zNzf · 2025-10-30

**Soundness:** 3
**Presentation:** 2
**Contribution:** 2
**Rating:** 2
**Confidence:** 3

**Summary:**

The paper presents an approach based on subspace projection using public data, namely using gradients evaluated on synthetic data that is generated using a public model. The DP-SGD algorithm is modified so that the gradients ar first projected in a certain way using these public gradients, the noise is added only to the small-dimensional objects which are then afterwards projected back to the high-dimensions. Experiments support the benefits of this approach.

**Strengths:**

- Interesting and a novel approach of using synthetic data to generated gradient projection bases. While there are lot existing works considering the projection of DP-SGD with public bases to reduce the amount of added noise, I am not aware of this specific approach in the literature, i.e., using public models to generate synthetic data for this task.

**Weaknesses:**

- the work overlooks several relevant and important prior studies, e.g., the work by [Gu et al., 2025](https://arxiv.org/pdf/2303.01256) and many others works that consider adding noise only to projected DP-SGD gradients. Existing works study different aspects of these methods. E.g., Gu et al. provide tools for privately estimating which public basis is most suitable for private gradients, an aspect highly relevant to the current paper’s methodology, yet not discussed at all.

- It remains unclear, why is the projection carried out the way it is done in Alg. 1. In  many existing projection methods the projection is carried out using orthonormal bases which is arguably much more stable alternative. There are no comparisons.

- The scales in the experiments are very narrow: the differences in the performance are in absolute sense small (look e.g. at Fig. 6,7,8). As the contribution is purely heuristical, the experimental part should be stronger and done much more carefully.

- No theoretical support for the proposed method.

**Questions:**

In  many existing projection methods the projection is carried out using orthonormal bases which is arguably much more stable alternative. Why this version?

How do you know which part of the generated synethic data is useful for the gradient projection? Why aren't there methods to filter out the most useful public data / gradients?

---

> ### Author Response · Authors · 2025-11-17
> **Response to Reviewer zNzf (1/4)**
>
> We thank our reviewer for the time and effort dedicated to evaluating our work, as well as for the constructive comments and suggestions. Our detailed responses are provided below. We have also revised the paper to further improve clarity and have highlighted all changes in blue.
>
> **Response to [W1] Related work discussion on noise added to projected gradients:**
>
> Thanks for pointing this out. First, we need to clarify that, ***we have already included the discussion of one of these papers*** of Yu et al. [1] in Section 4.2 **2) Existing Key Challenge** [C2] (see line 192 in the original paper and line 194 in the revised version), which forms an essential part of the foundational work leading to the paper you mentioned. We have also added the relevant baseline GEP from Yu et al. [1] in our revised paper.
>
> Second, while the paper you mentioned (Gu et al.) offers valuable insights, particularly in contributing to privately estimating which public basis best aligns with private gradients, its focus diverges from the goals of our work. Their method centers on *selecting an appropriate public dataset as a projection basis under differential privacy*, whereas in our work we ***do not rely on any public datasets*** (Section 1, lines 85 to 86) and ***do not include dataset selection*** as part of the projection mechanism.
>
> Instead, we focus on generating better synthetic datasets to serve as a projection subspace during DP fine-tuning with gradient projection and on empirical observations into finding a better place for DP noise injection. In this context, Yu et al. [1], who introduced the GEP gradient projection approach (***already included and discussed in our paper***), represents a more directly relevant line of prior work.
>
> That said, we appreciate the reviewer bringing Gu et al. to our attention. We are happy to include it in the revised version (see line 195 to 196 therein) to further strengthen the completeness and breadth of our related-work discussion.
>
> **Response to [W2] and [Q1] The connection and comparison between PE-SGD and orthonormal bases projection:**
>
> First, existing works regarding gradient projection using orthonormal bases like PDP-SGD **have already been included and compared** in our original paper (see Table 1, 2 and Figure 5, 6).
>
> What’s more, ***we have also already explained their relationship with a proof*** (see Section 4.2 line 175 to 179 and Appendix B.3 in our original paper; see also Section 4.2, line 179 to 181 and Appendix B.3 in the revised version) that ***these methods are mathematically equal to our method*** if they have not used the top-$k$ eigenspace but the whole space for their projection subspace.
> In the revised version, we additionally include GEP [1] in our experimental comparisons and prove that its orthonormal projection step (excluding its gradient-residual term) is mathematically equivalent to PE-SGD when using the full eigenspace rather than the top-$k$ components.
>
> Last but not least, we would like to clarify that, ***we have already explained the reason why we carried out the projection in the way that is done in Alg. 1*** in Section 4.2 **1) Gradient Projection**. A step by step mathematical deduction is presented therein from Eq. (1) to (3).
>
> ---
>
> [1] Yu, Da, et. al. "Do not let privacy overbill utility: Gradient embedding perturbation for private learning." ICLR2021.

---

> ### Author Response · Authors · 2025-11-17
> **Response to Reviewer zNzf (2/4)**
>
> **Response to [W3] Narrow experimental scale:**
>
> If “narrow experimental scale” points to the “seemingly minor change in loss and acc change for different parameters”, our response are the following:
>
> First, for Figure 6, we mainly focus on **limited private data and tight privacy budget** setting (see Section 1 and line 78 to 79 in our paper) where only a few hundreds of private samples are available and the standard DP-SGD typically struggles. This reflects practical constraints in real-world applications such as clinical domains [2]. In this context, increasing the number of private samples to $4,000$ already exceeds the scale relevant to our problem setting, yet even at this relatively large size, PE-SGD continues to outperform all baselines (see Table 1 in our paper). Moreover, because the case of $M=400$ already provides a clear performance separation between PE-SGD and other methods (see Table 1), exploring even smaller number of available private samples ($M$) values is not strictly necessary for demonstrating our method’s effectiveness. Nevertheless, to further address the reviewer’s concern, we conducted additional experiments with $M=100$. Under this more challenging regime, PE-SGD achieves a loss of $2.8428$ and accuracy of $0.4564$, compared with DP-SGD’s significantly worse performance with loss of $3.3329$ and accuracy $0.4374$. The strongest competing baseline, PDP-SGD, also underperforms PE-SGD, with a loss of $2.9622$ and accuracy of $0.4507$. These additional results further reinforce the robustness and soundness of our experimental conclusions within the problem setting of interest.
>
> Second, for Figure 7, like shown in Figure 12 in our original paper in Appendix D.3 (Figure 13 in Appendix D.3 in the revised version), varying $\epsilon$ across a wide range (from $1.0$ to $8.0$) has only a limited impact on the final performance for most methods. Moreover, although DP-SGD exhibits a larger performance variation, it requires a substantially larger differential privacy budget $\epsilon=8.0$ to reach a performance level that PE-SGD already achieves under a tight differential privacy budget $\epsilon=1.0$. This further demonstrates the superiority of PE-SGD in the setting of limited private samples and tight privacy budget. Therefore, the seemingly “narrow performance range’’ actually reflects a huge performance difference.
>
> Third, for Figure 8, it compares PE-SGD against two noise-placement variants across three datasets, each evaluated with three random seeds. While the absolute differences for individual datasets may appear small, the consistent ranking of methods across all datasets provides strong evidence for the superiority of the proposed noise placement strategy (inject on gradient coefficients).
>
> Overall, we believe that our experimental evaluation is thorough and strong, given its breadth and the inherent properties of PE-SGD and the baseline methods. If our reviewer still feels that the experiments fall short, we would greatly appreciate ***more specific and detailed*** guidance on which aspects are considered insufficient, rather than broad or general statements.
>
> ---
>
> [2] Soni, Sarvesh, and Dian Demner-Fushman. “ArchEHR-QA: BioNLP at ACL 2025 Shared Task on Grounded Electronic Health Record Question Answering.” Evaluation 11, no. 11.00: 7-80.

---

> ### Author Response · Authors · 2025-11-17
> **Response to Reviewer zNzf (3/4)**
>
> **Response to [W4] Theoretical support for the proposed PE-SGD:**
>
> We thank our reviewer for pointing this out. However, since we are mainly focusing on verifying the empirical usefulness of PE-SGD in this work, which should be treated as a start of exploiting evolutionary gradient projection subspaces, providing theoretical analysis can be an interesting future step.
>
> First, following the assumptions in PE [3] for evolutional DP synthetic data generation, if we assume that generated synthetic samples are never discarded (i.e., the synthetic dataset *grows* across iterations), then it is straightforward that the projection error under our evolutionary strategy cannot be worse than methods that rely on a fixed synthetic dataset: as the synthetic dataset expands, it spans a richer gradient subspace; thus its projection coverage will not get worse along the training progress.
>
> What’s more, we believe that **deriving a concrete theoretical convergence analysis is worth another individual paper*** and is significantly more challenging than that for PE. Our reasons are as follows.
>
> - **Dynamic optimization target.** In PE, the optimization target, the private samples, remains fixed across all iterations. In PE-SGD, the optimization target, the private gradients, varies across iterations, making the analysis substantially more complex.
>
> - **Synthetic gradients are also dynamic.** The gradients of the synthetic dataset depend on the current model parameters. Thus, even if the synthetic dataset was fixed, the induced gradients would *still* change over iterations. This stands in contrast to PE, where synthetic samples, and hence their quality assessment criteria, remain stable across iterations.
>
> - **Final convergence must be measured on model loss, not just subspace alignment.** Unlike PE, where the optimization target is to align synthetic samples with private samples, PE-SGD aims to ensure convergence of the fine-tuned model’s loss, a goal that is related to, but not directly equivalent to, aligning synthetic and private gradients. This adds an additional layer of complexity on top of the challenges already present. Although the Gradient Subspace Distance (GSD) bounds projection error (Lemma 4.2 in [4]) which further bounds the excess risk of public-data-assist (or synthetic-data-assisted) private machine learning algorithm like [5] (Theorem 4.1 in [4]) per iteration, and the ordering of GSD of several synthetic datasets with respect to the same private dataset persists across models and iterations, the synthetic dataset in PE-SGD evolves dynamically. This dynamicness makes deriving GSD-based guarantees more analytically difficult for the whole training process.
>
> In summary, the requirements for theoretical analysis of PE-SGD are more challenging than those for PE. Given that the theoretical analysis of PE alone already warrants a standalone paper [6], and that the analysis of PE-SGD is even more involved, we believe that a full theoretical treatment of PE-SGD is substantial enough to merit a separate publication.
>
> Nevertheless, we provide empirical evidence guided by Lemma 4.2 in [4], which states that the approximation error is bounded by the Gradient Subspace Distance (GSD). We compare the GSD of evolutionary vs. non-evolutionary synthetic gradients using our projection method in Table (G). The results show that evolutionary updates achieve consistently lower GSD across most iterations, supporting the claim that evolution improves subspace approximation and therefore yields better fine-tuning performance.
>
> Table (G). Gradient Subspace Distance (GSD) of the synthetic and private gradient subspace at each iteration. Qwen2.5-3B-Instruct is used with $M=400,\beta=0.2,T=10,N=200$.
> |Iteration|PubMed|PubMed|Congressional Speech|Congressional Speech|bioRxiv|bioRxiv |
> |:-:|:-:|:-:|:-:|:-:|:-:|:-:|
> || w/ evolution | w/o evolution | w/ evolution | w/o evolution | w/ evolution | w/o evolution |
> |1 | 6.17 | 6.17 | 6.58 | 6.58 | 6.55 | 6.55 |
> |2 | 6.19 | 6.25 | 6.62 | 6.64 | 6.66 | 6.66 |
> |3 | 6.17 | 6.17 | 6.64 | 6.67 | 6.68 | 6.69 |
> |4 | 5.60 | 5.75 | 6.70 | 6.66 | 6.65 | 6.74 |
> |5 | 6.31 | 6.27 | 6.71 | 6.71 | 6.80 | 6.82 |
> |6 | 6.44 | 6.46 | 6.65 | 6.77 | 6.80 | 6.85 |
> |7 | 6.59 | 6.51 | 6.68 | 6.78 | 6.84 | 6.86 |
> |8 | 6.56 | 6.59 | 6.72 | 6.74 | 6.82 | 6.84 |
> |9 | 6.49 | 6.61 | 6.66 | 6.71 | 6.81 | 6.82 |
> |10| 6.44 | 6.55 | 6.62 | 6.68 | 6.78 | 6.79 |
>
>  **Last, despite the difficulty, we provide a simple theoretical proof that can be a starting point in the following response.**
>
> ---
> [3] Lin, Zinan, et al. "Differentially Private Synthetic Data via Foundation Model APIs 1: Images." ICLR 2024.
>
> [4] Gu, Xin, et al. "Choosing public datasets for private machine learning via gradient subspace distance." IEEE SaTML 2025.
>
> [5] Yingxue, Zhou, et al. “Bypassing the Ambient Dimension: Private SGD with Gradient Subspace Identification.” ICLR 2021.
>
> [6] González, Tomás et al. "Private Evolution Converges." arXiv 2025.

---

> > ### Author Response · Authors · 2025-11-28
> > **A Theoretical Analysis on the Convergence Rate of PE-SGD (2/2)**
> >
> > ## Proof
> >
> > To prove **Theorem 1**, we first prove **Lemma 1** as follows.
> >
> > **Lemma 1 (VARIATIONAL_API Improvement on each group)**
> > Let $x_i$ be a synthetic sample, with $L-1$ samples $x_{i,l}, l=1,\cdots, L-1$ generated from the `VAIRATIONAL_API` using $x_i$ as the seed sample, i.e. $x_{i,l}$ is a variant of $x_i$. Denote $x_i$ as $x_{i,0}$ for clarity. Let $z_{i,l}$ be the gradient projection coefficient of $x_{i,l}$, and $v_{i,l} = z_{i,l}+n_{i,l}$ be the coefficient with DP noise where $n_{i,l}\sim \mathcal{N}(0,\sigma^2)$. Let $k_i=\arg\max_{0\leq l\leq L-1}v_{i,l}$ be the index of the selected sample. We have that with high probability, the selected sample improves the projection coefficient:
> >
> > $$
> > \Pr(|z_{i,k_i}| \geq |z_{i,0}|)   \geq 1-\exp\left(-\frac{(L-1)}{2}\cdot \Phi(\zeta/2\sigma)\right)
> > $$
> >
> >
> > ***Proof***
> >
> > Without loss of generality, we assume that $z_{i,0}>0$. For variants $j$ such as $z_{i,j}=z_{i,0}+\zeta$, we have
> > $$
> > \Pr(v_{i,j}< v_{i,0})\leq 1 -\Phi(\zeta/2\sigma)
> > $$
> > where $\Phi$ is the CDP of $\mathcal{N}(0,1)$.
> >
> > As the $L-1$ variants generated using the `VARIATIONAL_API` are independent, we have
> >
> > $$
> > \Pr(|z_{i,k_i}| \leq |z_{i,0}|)
> > \leq \Pr\left(\forall 1\leq l<L:(z_{i,l} = z_{i,0}-\zeta) \text{ or } (z_{i,l} = z_{i,0}+\zeta \text{ and } v_{i,l} < v_{i,0})\right)
> > $$
> > $$
> > = \left(\frac{1}{2} + \frac{1}{2}(1-\Phi(\zeta/2\sigma))\right)^{L-1}=\left(1-\frac{1}{2}\cdot \Phi(\zeta/2\sigma)\right)^{L-1}\leq \exp\left(-\frac{(L-1)}{2}\cdot \Phi(\zeta/2\sigma)\right).
> > $$
> >
> > The last inquality holds as $1-x\leq e^{-x}$ (after using first order Talyer expension). Therefore,
> > $$
> > \Pr(|z_{i,k_i}| \geq |z_{i,0}|)  \geq 1-\exp\left(-\frac{(L-1)}{2}\cdot \Phi(\zeta/2\sigma)\right).
> > $$
> >
> > This marks the end of proof for **Lemma 1**.
> >
> > ***Proof of Theorem 1***
> >
> > Since each group of variations is operated independently, we have
> > $$
> > \Pr(\sum_{i=1}^{\hat N} z_{s_i}^2 \geq \sum_{i=1}^{\hat N} z_i^2) \geq \Pr(\forall 1\leq i \leq \hat{N}: |z_{i,k_i}| \geq |z_{i,0}|) = \left[1-\exp\left(-\frac{(L-1)}{2}\cdot \Phi(\zeta/2\sigma)\right)\right]^{N/L}
> > $$
> >
> > $$
> > \geq 1-\frac{N}{L}\cdot \exp\left(-\frac{(L-1)}{2}\cdot \Phi(\zeta/2\sigma)\right).
> > $$
> >
> > This marks the end of the proof for **Theorem 1**.

---

> ### Author Response · Authors · 2025-11-17
> **Response to Reviewer zNzf (4/4)**
>
> **Response to [Q2] Synthetic sample evolution guidance and synthetic data filtration:**
>
> The gradient projection coefficient $z$ quantifies the contribution of each synthetic gradient to the final projected gradient. Therefore, synthetic samples corresponding to larger absolute value entries in $z$ naturally serve as more influential, and therefore more useful, contributors to the gradient projection.
>
> In fact, like ***what we have already discussed in Section 4.2 **3) Our solutions** [S1]*** (lines 203 to 205 in our original paper and lines 205 to 207 in the revised version), our evolutionary mechanism explicitly uses the absolute value of each element in $z$ to guide the selection of synthetic samples. We retain synthetic samples with larger $|z|$ as seed samples (samples that are used for generating new synthetic samples for the next training iteration) to encourage the generation of samples that have strong projection onto the private sample gradient subspace. We discard the non-selected samples in the above process and only keep the selected useful samples. From this perspective, our method ***have already incorporated synthetic sample filtering*** process. Moreover, $z$ can also be viewed as a “soft” filtration of synthetic sample gradients, as samples are weighted by the strength of their projection onto the private gradient subspace, with less perpendicular samples receiving higher weights.

---

> ### Author Response · Authors · 2025-11-28
> **A Theoretical Analysis on the Convergence Rate of PE-SGD (1/2)**
>
> We provide a convergence analysis below by demonstrating that "synthetic sample (gradient subspace) evolution improves the convergence of differentially private gradient projection algorithms with high probability".
>
> First, we need to clarify that, the key point of this proof is substantially different from prior works on DP gradient projection (e.g., PDP-SGD [1], the method you mentioned, and GEP [2]).
> Those works implicitly assume that all synthetic sample gradients follow the same distribution as the private gradients. However, as we empirically demonstrate in Table (G) in our previous respons (the "overlap" metric between synthetic and private gradient subspace is far smaller than $50$, which indicates that the two subspaces are identical), this assumption does not hold in practice.
> In contrast, the goal of PE-SGD is precisely to push the gradient subspace of synthetic samples to align with that of the private samples. Therefore, our proof focuses on characterizing the gradient alignment of evolution.
>
> More specifically, Theorem 3.3 in GEP [2] shows that the convergence rate is positively correlated with the average squared norm of the gradient-projection residual term. Thus, the central goal of our proof is to show that evolution reduces the upper bound of this average squared residual term with high probability, which directly improves the convergence behavior under the DP constraints.
>
>
> ## Notations and Assumptions
>
> We assume that $N$ is a multiple of $L$, and we have $\hat{N}= N/L$ synthetic samples, each with corresponding gradient $g_i\in\mathbb{R}^{p \times 1}$. Following Algorithm 1 of PE-SGD in the paper, after using `VARIATIONAL_API` for next generation synthetic dataset creation, we get another $\hat{N}(L-1)$ samples, with corresponding gradients ${\{g_i\}}_{i=\hat{N}+1}^N$. Let $h\in\mathbb{R}^{p \times 1}$ be the averaged gradient from the private samples.
>
> We make the following assumptions:
> * We use the clipping method to bound the sensitivity (Section 5.4), and assume that the $\ell_2$ norm of the gradient projection coefficients from each private samples are below the clipping threshold. This way, we can focus our analysis on the impact from the added DP noise.
> * We assume that the synthetic gradients have $\ell_2$ norm 1 (i.e., $\|g_i\|=1, \forall i=1,\cdots,N$) and are orthorgonal to each other (i.e., $<g_i,g_j>=0, \forall i\not=j$).
> * Let $z_i=(g_i^\top g_i)^{-1} g_i^\top h=\frac{g_i^\top h}{g_i^\top g_i}=\frac{g_i^\top h}{\|g_i\|^2}=g_i^\top h$ be the clean gradient projection coefficient, and $|z_i|$ is lower bounded by $\zeta>0$. We assume the variant sample $x_{i'}$ of $x_i$ created using the `VARIATIONAL_API` satisfies $z_{i'}=z_i+\zeta$ with probability 1/2 and $z_{i'}=z_i-\zeta$ with probability 1/2. This represents the setting where the `VARIATIONAL_API` could either improve or hurt the projection with the same probability.
> * To select the samples for the next iteration, we add i.i.d. DP noise $\nu\sim\mathcal{N}(0, \sigma^2)$ to the projection coefficient $z_i$, and select the samples with the top noisy coefficient from each variation group. We assume that the index of the selected samples are $s_1,...,s_{\hat{N}}$.
>
> ## Theorem 1 (High-Probability Improvement from Gaussian-Perturbed Variants)
> With a large possibility ($> 1-\tau$), the selected $\hat{N}$ samples have a sum of squared projection coefficients **not smaller** than that of the original $\hat{N}$ samples:
>
> $$
> \Pr(\sum_{i=1}^{\hat N} z_{s_i}^2 \geq \sum_{i=1}^{\hat N} z_i^2) \geq 1-\tau.
> $$
>
> where $\tau=\frac{N}{L}\cdot \exp\left(-\frac{(L-1)}{2}\cdot \Phi(\zeta/2\sigma)\right)$, and $\Phi$ is the CDP of $\mathcal{N}(0,1)$.
>
> ---
>
> ## Discussion
> This theorem (result) directly implies that the norm of the projected gradients  $\left\|\sum_{i=1}^{\hat{N}} z_i g_i\right\| = \sqrt{\sum_{i=1}^{\hat{N}} z_i^2}$  increases with high probability across iterations.  Consequently, the norm of the residual gradients on the selected $\hat{N}$ samples—an upper bound on the residual gradients across all $N$ samples—decreases with high probability over iterations.
>
> ---
>
> [1] Yingxue, Zhou, et al. “Bypassing the Ambient Dimension: Private SGD with Gradient Subspace Identification.” ICLR 2021.
>
> [2] Yu, Da, et. al. "Do not let privacy overbill utility: Gradient embedding perturbation for private learning." ICLR2021.

---

### Official Review · Reviewer_FYzi · 2025-11-02

**Soundness:** 1
**Presentation:** 2
**Contribution:** 1
**Rating:** 2
**Confidence:** 5

**Summary:**

This paper proposes PE-SGD, a framework for differentially private (DP) training that combines gradient subspace projection with an evolving synthetic dataset. The authors argue that existing projection-based DP methods (e.g., PDP-SGD, GEP) suffer from (1) fixed projection subspaces that fail to adapt during training, and (2) unclear choices of where to inject DP noise. PE-SGD addresses these by (a) updating the projection subspace via synthetic data generated by a pre-trained model at each iteration, and (b) empirically identifying that injecting noise into the final projection coefficients yields better performance. Experiments on text fine-tuning tasks with three pre-trained models show moderate improvements over prior methods under tight privacy budgets.

**Strengths:**

1. The paper tackles a practically relevant problem of improving DP-SGD performance when private data is limited, and situates itself within a growing body of work on gradient subspace methods.

2. The algorithm is presented clearly, and the privacy analysis (Appendix B.2) formally ensures that the DP guarantee remains equivalent to standard DP-SGD.

3. The implementation details, reproducibility statement, and visualizations are carefully documented.

**Weaknesses:**

1. Lack of depth in problem diagnosis

The paper highlights two “overlooked” issues in prior work --- fixed projection subspaces and unclear noise placement --- but does not provide a fundamental analysis or theoretical insight into either:

- For the fixed subspace issue, the paper only shows that in one experimental setup, the ℓ₂ distance between the private gradient and the projected noisy gradient increases with training (Fig. 1). This is a narrow empirical observation, not a principled explanation. There is no discussion of why the subspace mismatch arises (e.g., due to model drift, curvature changes, or gradient manifold evolution), nor analysis of whether this holds in general or depends on the dataset, model, or training regime.

- For the noise placement issue, the paper claims prior work lacked rationale --- but it likewise fails to provide one. The finding that adding noise to the final projection coefficients works best is purely empirical, without any theoretical justification or conceptual understanding.

Hence, although the paper frames itself as addressing two conceptual gaps, it mostly reiterates them experimentally rather than resolving them.

2. Weak connection between motivation and experiments

The paper’s motivation centers on differentially private training, yet all experiments are limited to fine-tuning pre-trained models using LoRA. This is a much narrower and simpler regime than full DP training from scratch. Consequently, the experimental evidence does not convincingly demonstrate the method’s broader applicability to general DP optimization, nor its scalability to large models or high-dimensional gradient spaces.

3. Overstated claims of originality

Both key ideas --- evolving a projection subspace and empirically selecting a noise placement --- are relatively incremental extensions of existing work rather than conceptually new mechanisms. The “evolution” is implemented via synthetic data regeneration using a generative model API, but there is no evidence that this evolution is guided or optimal, beyond heuristic sample resampling. The connection between this process and gradient-space adaptation remains speculative.

4. Limited theoretical depth

There is no analysis of convergence, bias-variance trade-offs, or the statistical efficiency of evolving subspaces under DP noise. The privacy argument relies entirely on post-processing invariance, which sidesteps the deeper question of whether iterative synthetic data generation introduces new privacy risks or affects gradient alignment stability.

5. Experimental limitations

While quantitative results show small gains under ε = 1.0, improvements are modest and primarily on fine-tuning tasks. There is no ablation for computational cost or efficiency, nor experiments beyond text data. The results, though positive, do not convincingly demonstrate a substantial advance in the state of the art.

**Questions:**

1. Can the authors provide any theoretical intuition for why fixed projection subspaces fail — e.g., does the private gradient manifold evolve significantly during fine-tuning, or is it a data-distribution drift effect?

2. Is the proposed “evolutionary” process guaranteed to improve gradient alignment, or could it destabilize the subspace under noise?

3. How is the additional synthetic data generation cost handled? Does this make PE-SGD slower than DP-SGD or PDP-SGD?

4. Since experiments only involve fine-tuning with LoRA, can the method extend to full-model DP training?

5. Can the authors provide at least a conceptual analysis of why adding noise to the projection coefficients yields better performance than to the raw gradients?

---

> ### Author Response · Authors · 2025-11-17
> **Response to Reviewer FYzi (1/5)**
>
> We thank our reviewer for the time and effort dedicated to evaluating our work, as well as for the constructive comments and suggestions. Our detailed responses are provided below. We have also revised the paper to further improve clarity and have highlighted all changes in blue.
>
> **Response to [W1.1 and Q1] Why fixed projection subspace fail:**
>
> We thank the reviewer for raising this interesting point. After a detailed analysis of the principal angles and subspace overlap of private gradient subspaces across iterations, we believe that the key reason lies in the substantial drift of the private gradient subspace during fine-tuning.
>
> Specifically, we compute the principal angles [1] between the private gradient subspace at the initial iteration, $H^{(0)}$, and those at later iterations, $H^{(t)}$ for $t = 1,\dots,T-1$.
> To compute these angles, we first extract the top-$50$ orthonormal basis vectors of each subspace, yielding $E^{(0)}, E^{(t)} \in \mathbb{R}^{p \times 50}$. We then form $O = {E^{(0)}}^\top E^{(t)}$ and compute its singular values $s_j$ for $j=1,\dots,50$ via SVD.
> The principal angles follow as $\theta_j = \arccos(s_j)$, lying in $[0^\circ, 90^\circ]$. Larger angles indicate greater divergence between subspaces, with $90^\circ$ corresponding to orthogonality. The “overlap” metric [2] is further defined as $overlap=\sum_{j=1}^{50} s_j^2$, depicting the similarity of the two subspaces. Therefore, smaller principal angles and larger “overlap” values indicate better alignment and resemblance of the two subspaces.
>
> Results are included in Figure 11 in the Appendix in our revised paper and in Table (D) below. As shown in Table (D), the subspace “overlap” metric drops sharply across iterations for all three tasks (private datasets) and different fine-tune models. This consistent decline demonstrates that the private gradient subspace evolves substantially during fine-tuning, independent of the specific dataset or fine-tuning model. Similar results can be drawn from Figure 11 as well. These findings clearly suggest that relying on a fixed projection subspace is suboptimal, as it fails to track the changing private gradient geometry during training.
>
> Table (D). “Overlap” values of the private gradient subspace at initial iterations and that at all other later iterations. $M=400,\beta=0.2,L=2,T=10,N=200$ is applied.
> | Iteration | PubMed | Congressional Speech | Congressional Speech | bioRxiv |
> |:-:|:-:|:-:|:-:|:-:|
> |  | Qwen2.5 | Qwen2.5 | Llama3.2 | Qwen2.5 |
> | 1 | 5.03 | 5.07 | 10.02 | 6.58 |
> | 2 | 3.88 | 5.23 | 7.28 | 5.35 |
> | 3 | 2.94 | 3.82 | 7.35 | 4.98 |
> | 4 | 2.81 | 3.12 | 4.33 | 4.40 |
> | 5 | 2.64 | 3.53 | 3.83 | 3.54 |
> | 6 | 2.24 | 2.76 | 4.25 | 2.14 |
> | 7 | 2.19 | 1.89 | 3.49 | 2.37 |
> | 8 | 1.75 | 2.25 | 3.59 | 2.58 |
> | 9 | 1.85 | 2.03 | 2.88 | 1.98 |
>
>
> **Response to [W1.2] Our rationale of noise placement:**
>
> Our reviewer states that our paper does not offer a rationale for choosing the noise injection location. We would like to clarify that our work ***does provide empirical guidance for selecting the noise placement point***, and this empirical evidence is itself a valid and meaningful rationale, especially given that prior literature has not examined this question in depth.
>
> We believe the reviewer’s criticism may stem from a misunderstanding of our phrasing. Our paper ***does not claim that “prior works lacked rationale”***; rather, we state that “prior works do not reveal their rationales’’ (see Section 1, line 75; see also Section 4.2, lines 193 to 195 in the original paper and lines 196 to 198). The distinction is important: existing methods simply do not discuss or analyze noise placement choices, effectively overlooking the issue. In contrast, our work is the first to explicitly identify noise placement as an important design factor and to provide empirical observations that can guide future decisions on where to inject DP noise.
>
>
> ---
>
> [1] Gu, Xin, et al. "Choosing public datasets for private machine learning via gradient subspace distance." In 2025 IEEE Conference on Secure and Trustworthy Machine Learning (SaTML), pp. 879-900. IEEE, 2025.
>
> [2] Zhang, Sikai, et al. "Canonical-correlation-based fast feature selection for structural health monitoring." Mechanical Systems and Signal Processing 223 (2025): 111895.

---

> > ### Author Response · Authors · 2025-11-17
> > **Response to Reviewer FYzi (5/5)**
> >
> > **Response to [Q5] Conceptual analysis of why adding noise to projection coefficients performs better than other options:**
> >
> > We ***have already included*** this in Section 5.4, lines 435 to 439, and Table 4 in Appendix D.4 in the original paper (see also Section 5.4, lines 439 to 443, and Table 5 in Appendix D.4 in the revised paper). By observing the STD-to-Mean ratio and Min-to-Max ratio of column norms of projection coefficients (PE-SGD) and raw gradients (PE-SGD-NoisyRealGrad), we observe that the former exhibits smaller norm disparities. Consequently, normalization introduces less distortion when noise is added to the projection coefficients. This provides a plausible explanation for why adding noise to the projection coefficients yields better performance.

---

> > > ### Comment · Reviewer_FYzi · 2025-11-20
> > >
> > > Thanks for the authors clarification. Most of my concerns have been addressed. I have increased my score
> > >
> > > However, the only thing is my last point: the theoretical analysis on the convergence rate. As the authors mentioned in the paper, previous papers also consider the random projection such as (https://arxiv.org/pdf/2007.03813). But most of them have convergence rate while there is no in this paper. I will consider to increase my score if the author can provide the convergence rate and comparison with previous results from a theoretical perspective.

---

> ### Author Response · Authors · 2025-11-17
> **Response to Reviewer FYzi (2/5)**
>
> **Response to [W2 and Q4] Motivation and the applicability of PE-SGD for full-model DP training:**
>
> First, we would like to clarify that our work mainly focuses on **limited private data and tight privacy budget** setting (see Section 1, lines 78 to 79) where only a few hundred private samples are available and where standard DP-SGD falls short. This scenario is common in real-world applications such as clinical domains [3]. Our goal is to design a DP training paradigm that performs effectively under these constraints. Under such conditions, the “full DP training from scratch” suggested by the reviewer is neither feasible nor aligned with the setting our method is designed to address.
>
> Moreover, PE-SGD can definitely be applied to full-model DP training with large models and high-dimensional gradients. We ***have already discussed this*** in Section 4.2 **4) Feasibility Discussion** (line 244 to 246 in the original paper and lines 246 to 248 in the revised paper). Since per-sample gradient dot products in PE-SGD can be computed efficiently using off-the-shelf tools such as GhostSuite [4], PE-SGD remains practical even for large-scale models with high-dimensional gradient spaces.
>
>
> **Response to [W3] Seemingly overstated claim of originality:**
>
> The claim given by our reviewer that PE-SGD is merely “an incremental extension of existing work” is not substantiated, as no specific prior work is cited to support this statement, which limits the credibility of the criticism.
>
> Moreover, our contributions are substantial and multifaceted, lying in the following aspects:
> - **(1) Identifying the suboptimality of fixed projection subspaces.** We are the first to show that using a fixed synthetic dataset as a gradient projection subspace is suboptimal, as the approximation error (measured by the $\ell_2$ distance between the approximate and real private gradients) grows substantially over iterations as the training progresses (see Figure 1 in our paper).
> - **(2) Introducing an evolution mechanism guided by gradient contribution.** To solve the above issue, we propose to “evolve” the synthetic dataset (also, its induced synthetic gradient subspace) ***guided by how much contribution a sample provides to the projection in the current round*** (see Algorithm 1, line 10, as we use the projection coefficient $z$ as the seed dataset selection guidance).
> - **(3) Highlighting the importance of noise placement.** We are also the first to explicitly identify noise placement as a critical factor for performance in DP gradient projection methods. None of the previous works has ever shed light on this point.
> - **(4) Providing empirical guidance on choosing the noise injection place.** Although a full theoretical guidance of optimal noise placement is challenging and deserves another individual paper, we provide empirical evidence and analysis for this question and the reason why adding noise onto projection coefficient is the best among the tested possibilities (see Section 5.4, lines 435 to 439, and Appendix D.4 in the original paper; see also Section 5.4, lines 439 to 443, and Appendix D.4 in the revised paper).
>
> To sum up, we believe that our contribution is original and indeed not minor. Also, the judgement that our evolution is “not guided” obviously diverges from the fact. Since we use *each sample’s contribution to the gradient projection* as the evolution guidance, the connection between the evolution process and the gradient subspace adaptation is direct and obvious, not speculative.
>
> ---
>
> [3] Soni, Sarvesh, and Dian Demner-Fushman. “ArchEHR-QA: BioNLP at ACL 2025 Shared Task on Grounded Electronic Health Record Question Answering.” Evaluation 11, no. 11.00: 7-80.
>
> [4] Jiachen T Wang et al. “Data Shapley in One Training Run.” ICLR2025.

---

> ### Author Response · Authors · 2025-11-17
> **Response to Reviewer FYzi (3/5)**
>
> **Response to [W4] In depth theoretical analysis on a wide range of aspects and post-processing invariance of DP for synthetic data evolution:**
>
> For the first concern of adding “convergence, bias-variance trade-offs, and the statistical efficiency of evolving subspaces under DP noise”, we believe that these theoretical analyses deserve another individual theoretical work, which could be an interesting future work and is not the focus of this work.
> 1. First, following the assumptions in PE [5] for evolutional DP synthetic data generation, if we assume that generated synthetic samples are never discarded (i.e., the synthetic dataset *grows* across iterations), then it is straightforward that the projection error under our evolutionary strategy cannot be worse than methods that rely on a fixed synthetic dataset: as the synthetic dataset expands, it spans a richer gradient subspace; thus its projection coverage will not get worse along the training progress.
> 2. What’s more, we believe that *deriving a concrete theoretical convergence analysis is worth another individual paper* and is significantly more challenging than that for PE. Our reasons are as follows.
> - - **Dynamic optimization target.** In PE, the optimization target, the private samples, remains fixed across all iterations. In PE-SGD, the optimization target, the private gradients, varies across iterations, making the analysis substantially more complex.
> - - **Synthetic gradients are also dynamic.** The gradients of the synthetic dataset depends on the current model parameters. Thus, even if the synthetic dataset was fixed, the induced gradients would *still* change over iterations. This stands in contrast to PE, where synthetic samples, and hence their quality assessment criteria, remain stable across iterations.
> - - **Final convergence must be measured on model loss, not just subspace alignment.** Unlike PE, where the optimization target is to align synthetic with private samples, PE-SGD aims to ensure convergence of the fine-tuned model’s loss, a goal that is related to, but not directly equivalent to, aligning synthetic and private gradients. This adds an additional layer of complexity on top of the challenges already present.
>
> In summary, the requirements for theoretical analysis of PE-SGD are more challenging than those for PE. Given that the theoretical analysis of PE alone already warrants a standalone paper [6], and that the analysis of PE-SGD is even more involved, we believe that a full theoretical treatment of PE-SGD is substantial enough to merit a separate publication.
>
> Nevertheless, we provide empirical evidence guided by Lemma 4.2 in [1], which states that the approximation error is bounded by the Gradient Subspace Distance (GSD). We compare the GSD of evolutionary vs. non-evolutionary synthetic gradients using our projection method in Table (G). The results show that evolutionary updates achieve consistently lower GSD across most iterations, supporting the claim that evolution improves subspace approximation and therefore yields better fine-tuning performance.
>
> For the second concern, we believe this concern arises from a misunderstanding of the post-processing invariance property of Differential Privacy (DP). As formally stated in [7], any post-processing that does not access the private data *does not* incur any additional privacy loss. Therefore, generating new synthetic samples using the fine-tuned model, even iteratively, does not create new privacy risks, because the private data is never accessed during this stage.
>
> Table (G). Gradient Subspace Distance (GSD) of the synthetic and private gradient subspace at each iteration. Qwen2.5-3B-Instruct is used with $M=400,\beta=0.2,T=10,N=200$.
> |Iteration|PubMed|PubMed|Congressional Speech|Congressional Speech|bioRxiv|bioRxiv|
> |:-:|:-:|:-:|:-:|:-:|:-:|:-:|
> ||w/ evolution|w/o evolution|w/ evolution|w/o evolution|w/ evolution|w/o evolution|
> |1 | 6.17 | 6.17 | 6.58 | 6.58 | 6.55 | 6.55 |
> |2 | 6.19 | 6.25 | 6.62 | 6.64 | 6.66 | 6.66 |
> |3 | 6.17 | 6.17 | 6.64 | 6.67 | 6.68 | 6.69 |
> |4 | 5.60 | 5.75 | 6.70 | 6.66 | 6.65 | 6.74 |
> |5 | 6.31 | 6.27 | 6.71 | 6.71 | 6.80 | 6.82 |
> |6 | 6.44 | 6.46 | 6.65 | 6.77 | 6.80 | 6.85 |
> |7 | 6.59 | 6.51 | 6.68 | 6.78 | 6.84 | 6.86 |
> |8 | 6.56 | 6.59 | 6.72 | 6.74 | 6.82 | 6.84 |
> |9 | 6.49 | 6.61 | 6.66 | 6.71 | 6.81 | 6.82 |
> |10| 6.44 | 6.55 | 6.62 | 6.68 | 6.78 | 6.79 |
>
> ---
>
> [1] Gu, Xin, et al. "Choosing public datasets for private machine learning via gradient subspace distance." IEEE SaTML 2025.
>
> [5] Lin, Zinan, et al. "Differentially Private Synthetic Data via Foundation Model APIs 1: Images." ICLR 2024.
>
> [6] González, Tomás et al. "Private Evolution Converges." arXiv 2025.
>
> [7] Cynthia, Dwork et al. “The Algorithmic Foundations of Differential Privacy.” Foundations  and Trends in Theoretical Computer Science 2014.

---

> ### Author Response · Authors · 2025-11-17
> **Response to Reviewer FYzi (4/5)**
>
> **Response to [W5] Computational comparison and experiments beyond text:**
>
> For comparisons on computational and cost of PE-SGD with other baseline methods, we ***have already included them*** in Table 8 in Appendix D.10 in our original submission (Table 11 in Appendix D.12 in revised version). Our reviewer may have overlooked this.
>
> For the requirement of “experiments beyond text” from our reviewer, multiple prior works on DP synthetic data only conduct experiments on one data modality respectively [3,8,9,10]. Therefore, given that we have already tested three diverse datasets using three different language models, we believe that our experiments are adequate enough for a top-tier conference.
> Moreover, we sincerely ask our reviewer to kindly reread our title “PE-SGD: Differentially Private Deep Learning via Evolution of Gradient Subspace ***for Text***” where we have clarified that we focus on the modality of text. Therefore, we think that it’s not proper for our reviewer to require experiments ***“beyond text”***.
>
>
> **Response to [Q2] Verification in alignment improvement through evolving synthetic gradient subspace:**
>
> We first state our conclusion: evolution brings better gradient subspace alignment compared to using fixed synthetic subspaces.
>
> This conclusion is drawn from observations into the “overlap” metric of synthetic gradient subspace and private gradient subspace across iterations, i.e. the “overlap” between $G^{(t)} \in \mathbf{R}^{p \times N}$ and $H^{(t)} \in \mathbb{R}^{p \times \tilde{M}}$.
> The “overlap” metric [2] is defined as $overlap=\sum_{j=1}^{50} s_j^2$, with $s_j, j=1,...,50$ being the singular values of $O={E_G^{(t)}}^T E_H^{(t)}$ and $E_G^{(t)}, E_H^{(t)}$ being the top-$50$ orthormonal basis of $G^{(t)}, H^{(t)}$ respectively [1].
>
> Like shown in Table (E), when a fixed synthetic gradient subspace is applied (“w/o evolution” columns), the “overlap” value drops by a larger extent compared to using evolved synthetic gradient subspaces. Our evolutionary mechanism helps to slow this process.
>
> Table (E). “Overlap” metric of the synthetic and private gradient subspace at each iteration. Qwen2.5-3B-Instruct is used with $M=400,\beta=0.2,T=10,N=200$.
> | Iteration | PubMed | PubMed | Congressional Speech | Congressional Speech | bioRxiv |bioRxiv |
> |:-:|:-:|:-:|:-:|:-:|:-:|:-:|
> |  | w/ evolution | w/o evolution | w/ evolution | w/o evolution | w/ evolution | w/o evolution |
> | 1 | 11.87 | 11.87 | 6.69 |6.69 | 7.11 | 7.11 |
> | 2 |11.64 | 9.88 | 5.80 | 5.59 | 5.56 | 5.58 |
> | 3 | 11.93 | 9.41 | 4.88 | 5.52 | 5.39 | 5.29 |
> | 4 | 6.36  | 8.89 | 5.06 | 4.97 | 5.77 | 4.59 |
> | 5 | 8.72 | 7.64 | 4.78 | 4.56 | 3.75 | 3.46 |
> | 6 | 8.10 | 7.54 | 5.74 | 4.14 | 3.71 | 3.10 |
> | 7 | 6.54 | 6.37 | 5.38 | 3.98 | 3.13 | 2.91 |
> | 8 | 6.88 | 6.51 | 4.71 | 4.51 | 3.40 | 3.18 |
> | 9 | 7.81 | 6.30 | 5.67 | 5.00 | 3.68 | 3.51 |
> |10| 8.57 | 6.85 | 6.16 | 5.27 | 3.89 | 3.68 |
>
> **Response to [Q3] Handeling the generation cost:**
>
> Thank you for raising this point. While evolving the synthetic dataset across iterations does introduce additional computational overhead, the required amount of synthetic data per iteration is small: around 200 samples per iteration when using API fold = 2 (i.e., only around 100 newly generated samples each iteration), as shown in Fig. 9(c). Consequently, the added cost remains modest and manageable in practice. This is further supported by the empirical generation times for 100 synthetic samples using Qwen2.5-3B-Instruct, reported in Table (F). We have also added this into our revised version in Appendix D.12.
>
> Table (F). Time costs (seconds, s) for generating 100 synthetic samples with Qwen2.5-3B-Instruct model using `RANDOM_API` and `VARIATIONAL_API` on 3 tasks. Tested with 80G-A100.
> |  | RANDOM_API |  VARIATIONAL_API |
> |:-|:-:|:-:|
> |PubMed| 26s | 41s |
> |Congressional Speech| 24s | 36s |
> |bioRxiv| 42s | 62s |
>
> ---
>
> [1] Gu, Xin, et al. "Choosing public datasets for private machine learning via gradient subspace distance." IEEE SaTML 2025.
>
> [2] Zhang, Sikai, et al. "Canonical-correlation-based fast feature selection for structural health monitoring." Mechanical Systems and Signal Processing 223 (2025): 111895.
>
> [3] Lin, Zinan, et al. "Differentially Private Synthetic Data via Foundation Model APIs 1: Images." ICLR 2024.
>
> [8] Xie, Chulin et al. "Differentially private synthetic data via foundation model apis 2: Text." ICML 2024.
>
> [9] Hou, Charlie et al. "PrE-Text: Training Language Models on Private Federated Data in the Age of LLMs." ICML 2024.
>
> [10] Zhang, Jianqing et al. "PCEvolve: Private Contrastive Evolution for Synthetic Dataset Generation via Few-Shot Private Data and Generative APIs." ICML 2025.

---

> ### Author Response · Authors · 2025-11-28
> **A Theoretical Analysis on the Convergence Rate of PE-SGD (1/2)**
>
> We thank our reviewer for the feedback. We provide a convergence analysis below by demonstrating that "synthetic sample (gradient subspace) evolution improves the convergence of differentially private gradient projection algorithms with high probability".
>
> First, we need to clarify that, the key point of this proof is substantially different from prior works on DP gradient projection (e.g., PDP-SGD [1], the method you mentioned, and GEP [2]).
> Those works implicitly assume that all synthetic sample gradients follow the same distribution as the private gradients. However, as we empirically demonstrate in Table (E) in our previous respons (the "overlap" metric between synthetic and private gradient subspace is far smaller than $50$, which indicates that the two subspaces are identical), this assumption does not hold in practice.
> In contrast, the goal of PE-SGD is precisely to push the gradient subspace of synthetic samples to align with that of the private samples. Therefore, our proof focuses on characterizing the gradient alignment of evolution.
>
> More specifically, Theorem 3.3 in GEP [2] shows that the convergence rate is positively correlated with the average squared norm of the gradient-projection residual term. Thus, the central goal of our proof is to show that evolution reduces the upper bound of this average squared residual term with high probability, which directly improves the convergence behavior under the DP constraints.
>
>
> ## Notations and Assumptions
>
> We assume that $N$ is a multiple of $L$, and we have $\hat{N}= N/L$ synthetic samples, each with corresponding gradient $g_i\in\mathbb{R}^{p \times 1}$. Following Algorithm 1 of PE-SGD in the paper, after using `VARIATIONAL_API` for next generation synthetic dataset creation, we get another $\hat{N}(L-1)$ samples, with corresponding gradients ${\{g_i\}}_{i=\hat{N}+1}^N$. Let $h\in\mathbb{R}^{p \times 1}$ be the averaged gradient from the private samples.
>
> We make the following assumptions:
> * We use the clipping method to bound the sensitivity (Section 5.4), and assume that the $\ell_2$ norm of the gradient projection coefficients from each private samples are below the clipping threshold. This way, we can focus our analysis on the impact from the added DP noise.
> * We assume that the synthetic gradients have $\ell_2$ norm 1 (i.e., $\|g_i\|=1, \forall i=1,\cdots,N$) and are orthorgonal to each other (i.e., $<g_i,g_j>=0, \forall i\not=j$).
> * Let $z_i=(g_i^\top g_i)^{-1} g_i^\top h=\frac{g_i^\top h}{g_i^\top g_i}=\frac{g_i^\top h}{\|g_i\|^2}=g_i^\top h$ be the clean gradient projection coefficient, and $|z_i|$ is lower bounded by $\zeta>0$. We assume the variant sample $x_{i'}$ of $x_i$ created using the `VARIATIONAL_API` satisfies $z_{i'}=z_i+\zeta$ with probability 1/2 and $z_{i'}=z_i-\zeta$ with probability 1/2. This represents the setting where the `VARIATIONAL_API` could either improve or hurt the projection with the same probability.
> * To select the samples for the next iteration, we add i.i.d. DP noise $\nu\sim\mathcal{N}(0, \sigma^2)$ to the projection coefficient $z_i$, and select the samples with the top noisy coefficient from each variation group. We assume that the index of the selected samples are $s_1,...,s_{\hat{N}}$.
>
> ## Theorem 1 (High-Probability Improvement from Gaussian-Perturbed Variants)
> With a large possibility ($> 1-\tau$), the selected $\hat{N}$ samples have a sum of squared projection coefficients **not smaller** than that of the original $\hat{N}$ samples:
>
> $$
> \Pr(\sum_{i=1}^{\hat N} z_{s_i}^2 \geq \sum_{i=1}^{\hat N} z_i^2) \geq 1-\tau.
> $$
>
> where $\tau=\frac{N}{L}\cdot \exp\left(-\frac{(L-1)}{2}\cdot \Phi(\zeta/2\sigma)\right)$, and $\Phi$ is the CDP of $\mathcal{N}(0,1)$.
>
> ---
>
> ## Discussion
> This theorem (result) directly implies that the norm of the projected gradients  $\left\|\sum_{i=1}^{\hat{N}} z_i g_i\right\| = \sqrt{\sum_{i=1}^{\hat{N}} z_i^2}$  increases with high probability across iterations.  Consequently, the norm of the residual gradients on the selected $\hat{N}$ samples—an upper bound on the residual gradients across all $N$ samples—decreases with high probability over iterations.
>
> ---
>
> [1] Yingxue, Zhou, et al. “Bypassing the Ambient Dimension: Private SGD with Gradient Subspace Identification.” ICLR 2021.
>
> [2] Yu, Da, et. al. "Do not let privacy overbill utility: Gradient embedding perturbation for private learning." ICLR2021.

---

> ### Author Response · Authors · 2025-11-28
> **A Theoretical Analysis on the Convergence Rate of PE-SGD (2/2)**
>
> ## Proof
>
> To prove **Theorem 1**, we first prove **Lemma 1** as follows.
>
> **Lemma 1 (VARIATIONAL_API Improvement on each group)**
> Let $x_i$ be a synthetic sample, with $L-1$ samples $x_{i,l}, l=1,\cdots, L-1$ generated from the `VAIRATIONAL_API` using $x_i$ as the seed sample, i.e. $x_{i,l}$ is a variant of $x_i$. Denote $x_i$ as $x_{i,0}$ for clarity. Let $z_{i,l}$ be the gradient projection coefficient of $x_{i,l}$, and $v_{i,l} = z_{i,l}+n_{i,l}$ be the coefficient with DP noise where $n_{i,l}\sim \mathcal{N}(0,\sigma^2)$. Let $k_i=\arg\max_{0\leq l\leq L-1}v_{i,l}$ be the index of the selected sample. We have that with high probability, the selected sample improves the projection coefficient:
>
> $$
> \Pr(|z_{i,k_i}| \geq |z_{i,0}|)   \geq 1-\exp\left(-\frac{(L-1)}{2}\cdot \Phi(\zeta/2\sigma)\right)
> $$
>
>
> ***Proof***
>
> Without loss of generality, we assume that $z_{i,0}>0$. For variants $j$ such as $z_{i,j}=z_{i,0}+\zeta$, we have
> $$
> \Pr(v_{i,j}< v_{i,0})\leq 1 -\Phi(\zeta/2\sigma)
> $$
> where $\Phi$ is the CDP of $\mathcal{N}(0,1)$.
>
> As the $L-1$ variants generated using the `VARIATIONAL_API` are independent, we have
>
> $$
> \Pr(|z_{i,k_i}| \leq |z_{i,0}|)
> \leq \Pr\left(\forall 1\leq l<L:(z_{i,l} = z_{i,0}-\zeta) \text{ or } (z_{i,l} = z_{i,0}+\zeta \text{ and } v_{i,l} < v_{i,0})\right)
> $$
> $$
> = \left(\frac{1}{2} + \frac{1}{2}(1-\Phi(\zeta/2\sigma))\right)^{L-1}=\left(1-\frac{1}{2}\cdot \Phi(\zeta/2\sigma)\right)^{L-1}\leq \exp\left(-\frac{(L-1)}{2}\cdot \Phi(\zeta/2\sigma)\right).
> $$
>
> The last inquality holds as $1-x\leq e^{-x}$ (after using first order Talyer expension). Therefore,
> $$
> \Pr(|z_{i,k_i}| \geq |z_{i,0}|)  \geq 1-\exp\left(-\frac{(L-1)}{2}\cdot \Phi(\zeta/2\sigma)\right).
> $$
>
> This marks the end of proof for **Lemma 1**.
>
> ***Proof of Theorem 1***
>
> Since each group of variations is operated independently, we have
> $$
> \Pr(\sum_{i=1}^{\hat N} z_{s_i}^2 \geq \sum_{i=1}^{\hat N} z_i^2) \geq \Pr(\forall 1\leq i \leq \hat{N}: |z_{i,k_i}| \geq |z_{i,0}|) = \left[1-\exp\left(-\frac{(L-1)}{2}\cdot \Phi(\zeta/2\sigma)\right)\right]^{N/L}
> $$
>
> $$
> \geq 1-\frac{N}{L}\cdot \exp\left(-\frac{(L-1)}{2}\cdot \Phi(\zeta/2\sigma)\right).
> $$
>
> This marks the end of the proof for **Theorem 1**.

---

### Official Review · Reviewer_eNfj · 2025-11-03

**Soundness:** 3
**Presentation:** 3
**Contribution:** 2
**Rating:** 6
**Confidence:** 3

**Summary:**

The paper proposes an improved subspace projection-based DP-SGD like training algorithm, with the idea of identifying the subspace of gradient using data generated by the model being trained itself. Experiments show the proposed algorithm can achieve better performance when privacy budget is limited where DP-SGD tends to fail.

**Strengths:**

1. The algorithm is built on a simple and neat idea of using samples generated by model itself to identify the gradient subspace, removing the need of a in-distribution public dataset and mitigate the potential issue of misaligned gradient subspace.
2. The algorithm achieves better empirical performance than baselines when epsilon is relatively small.

**Weaknesses:**

1. The proposed algorithm need additional computation to sample new dataset across iterations, which could make the algorithm relatively slow.
2. The algorithm performance may depends on the quality of generated data itself, which may imply the original model quality and capability performs a role here. There is a lack of discussion and ablation on this topic.

**Questions:**

1. Could the authors comment on how generated data quality and original model capability could affect final model performance?
2. Did authors visualized how the subspace of projection obtained from synthetic data compared with subspace of the model gradient? Also how does prompt template of data generation affect the subspace difference and final performance?

---

> ### Author Response · Authors · 2025-11-17
> **Response to Reviewer eNfj (1/2)**
>
> We thank our reviewer for the time and effort dedicated to evaluating our work, as well as for the constructive comments and suggestions. Our detailed responses are provided below. We have also revised the paper to further improve clarity and have highlighted all changes in blue.
>
> **Response to [W1] Efficiency of synthetic dataset evolution:**
>
> Thank you for raising this point. While evolving the synthetic dataset across iterations does introduce additional computational overhead, the required amount of synthetic data per iteration is small: around 200 samples per iteration when using API fold = 2 (i.e., only around 100 newly generated samples each iteration), as shown in Fig. 9(c). Consequently, the added cost remains modest and manageable in practice. This is further supported by the empirical generation times for 100 synthetic samples using Qwen2.5-3B-Instruct, reported in Table (A). We have also added this into our revised version in Appendix D.12.
>
> Table (A). Time costs (seconds, s) for generating 100 synthetic samples with Qwen2.5-3B-Instruct model using `RANDOM_API` and `VARIATIONAL_API` on 3 tasks. Tested with 80G-A100.
> |  | RANDOM_API |  VARIATIONAL_API |
> |:-|:-:|:-:|
> |PubMed| 26s | 41s |
> |Congressional Speech| 24s | 36s |
> |bioRxiv| 42s | 62s |
>
> ---
>
> **Response to [W2 and Q1] Generated Data Quality and Performance:**
>
> We thank our reviewer for pointing this out. Our responses are as follows which is also added to our revised version in Section 5.4 (highlighted in blue).
>
> First, there is no doubt that starting from a stronger base model generally yields better performance. The performance gap between using Qwen2.5-3B-Instruct and GPT-2 in our experiments (Table 1 and Figure 5 in the paper) already demonstrates this.
>
> What’s more, if one is concerned about the quality of the generated dataset and hopes to rely on a more powerful model for synthetic data generation, our PE-SGD naturally supports this setting. Our method can be easily extended to “generating synthetic samples using a stronger model, then performing DP fine-tuning on a weaker model using these synthetic samples” by simply freezing the generation model parameters in Algorithm 1. As shown in Table (B), this setting further improves the final model performance.
>
> In Table (B), we compare two settings: (i) generating synthetic data using the fine-tuned Qwen2.5-3B-Instruct itself, and (ii) generating using a more powerful model, namely Qwen2.5-7B-Instruct or GPT-4o-mini, while still fine-tuning the same Qwen2.5-3B-Instruct target model. Results clearly show that using the stronger generation model leads to consistent performance gains over using Qwen2.5-3B-Instruct alone.
>
> Last but not least, our reviewer might be concerned about the possible limitation that if the initial model is not good enough, the performance of our PE-SGD would be affected. However, this possible limitation is not redistributed to our PE-SGD, but also for all other DP fine-tuning methods like DP-SGD and projection based methods like PDP-SGD compared in our paper. Therefore, this should not be treated as a reason for rejecting our paper.
>
> Table (B) Fine-tuning Qwen2.5-3B-Instruct using synthetic samples generated by different LLMs. $M=400,\beta=0.2,N=200,T=10$ is applied.
> | Generation Model | $\epsilon$ | PubMed | PubMed | Congressional | Congressional | bioRxiv |  bioRxiv |
> |:-:|:-:|:-:|:-:|:-:|:-:|:-:|:-:|
> |  |  | Loss | ACC | Loss | ACC | Loss | ACC |
> | Qwen2.5-3B-Instruct | $1.0$ | 2.1990$\pm$0.0124 | 0.5435$\pm$0.0009 | 2.7532$\pm$0.0166 | 0.4577$\pm$0.0011 | 2.3178$\pm$0.0018 | 0.5095$\pm$0.0000 |
> | Qwen2.5-7B-Instruct | $1.0$ | 2.1879$\pm$0.0319 | 0.5480$\pm$0.0025 | 2.7152$\pm$0.0233 | 0.4615$\pm$0.0022 | 2.3042$\pm$0.0023 | 0.5139$\pm$0.0003 |
> | GPT-4o-mini | $1.0$ | 2.1862$\pm$0.0051 | 0.5480$\pm$0.0007 | 2.7219$\pm$0.0367 | 0.4618$\pm$0.0013 | 2.3013$\pm$0.0031 | 0.5142$\pm$0.0002 |
> | Qwen2.5-3B-Instruct | $\infty$ | 2.1839$\pm$0.0131 | 0.5436$\pm$0.0005 | 2.7468$\pm$0.0095 | 0.4579$\pm$0.0021 | 2.3165$\pm$0.0040 | 0.5097$\pm$0.0004 |
> | Qwen2.5-7B-Instruct | $\infty$ | 2.1794$\pm$0.0340 | 0.5493$\pm$0.0026 | 2.7104$\pm$0.0266 | 0.4620$\pm$0.0019 | 2.3015 $\pm$0.0030 | 0.5148$\pm$0.0004 |
> | GPT-4o-mini | $\infty$ | 2.1737$\pm$0.0089 | 0.5495$\pm$0.0007 | 2.7198$\pm$0.0359 | 0.4622$\pm$0.0013 | 2.3010$\pm$0.0024 | 0.5148$\pm$0.0002 |

---

> ### Author Response · Authors · 2025-11-17
> **Response to Reviewer eNfj (2/2)**
>
> **Response to [Q2] Visualization of synthetic and real gradient subspaces comparison with respect to different prompts:**
>
> We include a PCA visualization of the synthetic and real gradient subspaces of PE-SGD in the revised paper (see left column of Figure 14 (a),(c) and (e) in Appendix D.6 of our revised paper). These figures show that their alignment improves substantially over the course of training compared with the initial iteration, and is better than that of the fixed-projection baseline (“PE-SGD-FixSample” in Figure 14 (b), (d) and (f) in our revised paper). This provides clear evidence for the effectiveness of the evolving projection mechanism in PE-SGD.
>
> Moreover, to assess the impact of different prompt templates on data synthesis, we follow prior work [1] and report final model performance as well as FID, Precision, and Recall of the synthetic dataset in Table (C) on Congressional Speech task using Qwen2.5. Note that the FID, Precision and Recall metrics are computed using 200 randomly generated samples from the fine-tuned model after the whole training process. The first row corresponds to the prompt used in our main results (“Original”), which was designed following prior work [1]. “P-1” removes all tone options from the one-shot prompt for `VARIATIONAL_API`, while keeping the zero-shot prompt for `RANDOM_API` unchanged. “P-2” keeps only three tones in the one-shot prompt for `VARIATIONAL_API`, again with the `RANDOM_API` prompt unchanged relative to “Original.” “P-3” and “P-4” use prompts rewritten by ChatGPT, with “P-3” being more structured and “P-4” being more creativity-oriented (as judged by ChatGPT). Full prompt details are provided in Table 7 in Appendix D.5 in our revised paper.
>
> From Table (C), we observe that using different reasonable prompts has only a minor impact on final model performance (Loss and ACC). However, different prompts affect the Precision and Recall of the synthetic dataset with respect to the private dataset.
>
> On one hand, although less diversity inspiring prompts (“P-1” and “P-2” vs. “Original”) preserve final model performance and FID, they significantly reduce Precision, indicating an increasing number of diverging samples that are less correlated to the target private distribution. On the other hand, a more structured prompt (“P-3” vs. “Original”) improves Recall (covering more of the private sample space) but harms Precision (introducing more synthetic samples that deviate from the private distribution); while a more creative-inspiring prompt (“P-4”) performs worse overall than the others.
>
> Table (C) Comparison of final model performance, FID, Precision, Recall of the generated synthetic dataset using different prompts throughout PE-SGD training process with Qwen2.5-3B-Instruct.
> | Prompt | Loss | ACC | FID | Precision | Recall |
> |:-:|:-:|:-:|:-:|:-:|:-:|
> | Original | 2.7532$\pm$0.0166 | 0.4577$\pm$0.0011 | 0.1571$\pm$0.0056 | 0.9200$\pm$0.0537 | 0.5235$\pm$0.1032 |
> | P-1 | 2.7526$\pm$0.0136 | 0.4570$\pm$0.0012 | 0.1625$\pm$0.0121 | 0.8533$\pm$0.0153 | 0.5416$\pm$0.1860 |
> | P-2 | 2.7522$\pm$0.0336 | 0.4572$\pm$0.0012 | 0.1524$\pm$0.0048 | 0.8633$\pm$0.0681 | 0.5233$\pm$0.1015 |
> | P-3 | 2.7915$\pm$0.0493 | 0.4576$\pm$0.0025 | 0.1767$\pm$0.0135 | 0.7200$\pm$0.1609 | 0.6483$\pm$0.1703 |
> | P-4 | 2.8017$\pm$0.0995 | 0.4574$\pm$0.0020 | 0.2054$\pm$0.0097 | 0.6933$\pm$0.1320 | 0.5708$\pm$0.3621 |
>
> ---
>
> [1] Xie, Chulin et al. "Differentially private synthetic data via foundation model apis 2: Text." ICML 2024.

---

### Meta-Review · Area_Chair_haoT · 2025-12-25

**Summary:**

This paper proposes PE-SGD, which dynamically updates the non-private dataset based on the evolving model during training, thereby reducing the L2 distance between the approximate gradients and the real gradients. In addition, the paper systematically compares different noise injection strategies and finds that adding noise to the final projection coefficients yields more stable and superior performance than injecting noise at other stages.

The reviewers have received various scores: 6 from reviewers eNfj, v9H2  and 3 from FYzi, and zNzf.  The authors also provided very detailed response to the authors and also very detailed Rebuttal Summarization.   Reviewer  FYzi recommended rejection and raised various comments which, from my point of view, were well addressed by the authors. more importantly,   Reviewer  FYzi mentioned that they will consider to increase my score if the author can provide the convergence rate and comparison. Reviewer  FYzi  did not post a response after the authors successfully provided the convergence rate. I would consider the FYzi will at least not object to the acceptance of this submission since the authors addressed all the concerns raised by them.

the other Reviwer zNzf voted for rejection mentioned that the confidence score is low and they did not participate the discussion after the authors provided detailed feedback to their concerns.

Based on the above reasons, I concluded the proposed method  seems to be novel which supported by theory and experiments. The authors did a good job in the rebuttal which seems to be convincing to me.  I would recommend its acceptance. I strongly recommend the authors to address fully  the reivewers' comments in the final version.

**Reviewer Concerns:**

theoretical analysis was successfully addressed in the rebuttal. Generated synthetic sample quality impact and quality assessment were also addressed in the rebuttal with additional experiments relying on a more powerful LLM as the synthetic sample generation model (Reviewer eNfj) and reported several dataset quality measurement metrics including FID, Precision, Recall and Principle Angles etc.

**Reviewer Scores:**

Reviewer  FYzi mentioned that they will consider to increase my score if the author can provide the convergence rate and comparison. Reviewer  FYzi  did not post a response after the authors successfully provided the convergence rate. I would consider the FYzi will at least object to the acceptance of this submission since the authors addressed all the concerns raised by them.

---

### Decision · Program_Chairs · 2026-01-26

Accept (Poster)